# Partition to Evolve: Niching-enhanced Evolution with LLMs for Automated Algorithm Discovery

**Qinglong Hu**
Department of Computer Science
City University of Hong Kong
Hong Kong, China
qinglhu2-c@my.cityu.edu.hk

**Qingfu Zhang**
Department of Computer Science
City University of Hong Kong
Hong Kong, China
qingfu.zhang@cityu.edu.hk

## Abstract

Large language model-assisted Evolutionary Search (LES) has emerged as a promising approach for Automated Algorithm Discovery (AAD). While many evolutionary search strategies have been developed for classic optimization problems, LES operates in abstract language spaces, presenting unique challenges for applying these strategies effectively. To address this, we propose a general LES framework that incorporates feature-assisted niche construction within abstract search spaces, enabling the seamless integration of niche-based search strategies from evolutionary computation. Building on this framework, we introduce *PartEvo* (Partition to Evolve), an LES method that combines niche collaborative search and advanced prompting strategies to improve algorithm discovery efficiency. Experiments on both synthetic and real-world optimization problems show that PartEvo outperforms human-designed baselines and surpasses prior LES methods. In particular, on resource scheduling tasks, PartEvo generates meta-heuristics with low design costs, achieving up to 90.1% performance improvement over widely-used baseline algorithms, highlighting its potential for real-world applications.

## 1 Introduction

Algorithms are essential for solving real-world problems. As demands across domains become increasingly diverse and complex, Automated Algorithm Discovery (AAD) has emerged as a critical approach to improving the efficiency of algorithm development [1]. Techniques such as genetic programming, auto-regressive models, and reinforcement learning have successfully automated tasks like algorithm configuration and composition [2–4]. However, the automated generation of more sophisticated algorithms remains an ongoing pursuit.

Recent advancements in Large Language Models (LLMs) have provided transformative tools for algorithm generation, owing to their impressive capabilities in natural language understanding and code generation [5]. Among these, the LLM-assisted Evolutionary Search (LES) paradigm has shown great promise in AAD tasks. LES represents search objects (e.g., algorithms, code, heuristics) as individuals in an evolutionary framework, where LLMs, guided by specialized prompts, act as evolutionary operators to improve these individuals iteratively [6–9]. While early LES methods have demonstrated the feasibility and potential of LES in AAD, they often rely on oversimplified search mechanisms (e.g., greedy selection), which limit their efficiency.

In Evolutionary Computation (EC), search efficiency is enhanced through better exploration-exploitation trade-offs, which can be achieved via computational resource allocation techniques such as niching [10] and search space partitioning [11]. It is natural to consider adapting these established EC techniques to the LES to enhance algorithm discovery. However, applying them to LES poses new challenges due to the fundamental shift in the nature of the search space.

39th Conference on Neural Information Processing Systems (NeurIPS 2025).

With the integration of LLMs, the search space extends beyond traditional numerical or manually designed discrete spaces into language spaces [12]. Unlike traditional search spaces, language spaces lack explicit dimensionality and well-defined structures. Instead, they are implicitly shaped by the interaction between the LLMs and the specific task context. This abstraction complicates the adoption of advanced EC techniques. For example, in numerical domains, niches can often be defined using distance thresholds [13]. However, in language spaces, it is inherently challenging to compute distances between algorithms, which hinders the application of niche-based EC techniques.

To address these challenges, we present a practical pipeline for partitioning language search spaces and constructing niches, integrated into a general LES framework to improve search efficiency. This framework facilitates the seamless incorporation of niche-based EC techniques into LES, enabling more effective allocation of sampling resources (i.e., queries to LLMs) during the search process. Critically, it also establishes a methodological blueprint for incorporating a diverse range of advanced EC methods into future LES pipelines. Building on this foundation, we propose *PartEvo*, an LES method that combines advanced prompting strategies with effective EC techniques. PartEvo significantly improves search efficiency and excels in AAD tasks, particularly under limited sampling budgets. Our contributions can be summarized as follows:

- **A general LES framework for integrating advanced EC techniques:** We propose a general LES framework that incorporates feature-assisted abstract search space partitioning, enabling structured (non-random) niche construction for the algorithm discovery process. This framework allows LES to integrate niche-based EC techniques, enhancing the efficiency of sampling resource allocation in language search spaces.

- **PartEvo development:** We develop *PartEvo*, a novel LES method that combines verbal gradients with both local and global search strategies to discover high-performing algorithms. PartEvo exemplifies the seamless integration of EC techniques with LES, demonstrating significant performance and efficiency gains.

- **Comprehensive evaluation:** We evaluate PartEvo on AAD tasks over both synthetic and real-world optimization problems. It consistently outperforms human-designed meta-heuristic baselines and achieves substantial gains in algorithm discovery efficiency. Furthermore, extensive ablation studies validate the effectiveness of feature-assisted niche construction, advanced prompting strategies, and the integration of advanced EC techniques into LES.

## 2 Related works

**Automated algorithm discovery**  Automatic algorithm discovery automates the configuration, combination, or generation of algorithms tailored to specific problems, yielding significant improvements in efficiency and scalability [14, 4, 15]. It has become possible to automatically tune hyperparameters or combine different algorithmic components [16, 17]. Genetic programming [18, 19] provides an interpretable approach to algorithm design. Significant efforts have also been made to incorporate machine learning techniques [20–22, 3] into the automatic algorithm design process. More recently, pre-trained LLMs, with their extensive knowledge repositories, have shown considerable promise in the field of automated algorithm discovery [23, 24]. However, the full potential of LLMs for effective algorithm discovery remains largely untapped in current LES approaches.

**LLM-assisted evolutionary search**  EC is a powerful optimization paradigm inspired by the principles of natural evolution [25, 26]. The integration of LLMs into EC has led to significant advancements in areas such as code generation [27, 28] and text generation [29, 30]. Moreover, the combination of EC and LLMs, particularly through prompt engineering, has shown remarkable potential across various domains, including scientific discovery [6, 31], algorithmic component design [32, 33], reward function optimization [9, 34], and neural architecture search [35]. Recent studies have also investigated specialized individual encoding strategies [7] and verbal gradients [8] to further elicit the contextual understanding capabilities of LLMs during iterative evolutionary processes. Building on these developments, this work introduces a general framework incorporating niche-based EC techniques to improve the search efficiency of LES.

**Niching principles**  Niching is a widely used strategy in optimization, designed to improve efficiency by dividing the search space into smaller, more manageable subregions [36, 37]. It has shown strong

benefits in black-box optimization [11] and Bayesian optimization [38]. In AutoML, partitioning improves the construction of machine learning pipelines by reducing search time [39]. In numerical optimization, niche-based methods are often employed to avoid convergence to local optima in complex problems [10, 40]. Inspired by these successes, we seek to expand the concept of niching to language search spaces, integrating it into LES-driven algorithm discovery. This enables more structured and controllable allocation of sampling resources, leading to improved search efficiency.

## 3 Niching-enhanced LLM-assisted evolutionary search

### 3.1 General framework

In the LES process, the language search space is inherently defined when using an LLM for a specific task. A single search step involves selecting parent algorithms, embedding them into few-shot prompts, and querying the LLM to generate new candidate algorithms. This can be viewed as a sampling operation within the language search space. Thus, the LES process can be interpreted as iterative sampling in this abstract space facilitated by the LLM. Our goal is to partition this space into subregions (i.e., niches), enabling a structured allocation of sampling resources across the entire space that strikes a balance between exploration and exploitation, ultimately enhancing search efficiency.

A key challenge arises from the lack of explicit representations of language search spaces. Unlike numerical spaces, where niches can be defined via geometric or distance-based thresholds, language spaces do not have clear dimensionality or structure. Each sampled point corresponds to a complete text or a specific semantic expression, with no numerically defined "distance" between points. This implicit nature complicates the analysis and partitioning, limiting the application of evolutionary strategies that depend on structured subspaces for efficient searching.

We address this challenge by recognizing that each sampled point originates from this abstract search space. By analyzing the sampling results, we can indirectly infer and partition the original language space. To achieve this, we propose a feature-assisted partitioning pipeline that allows for non-random niche construction, making it possible to incorporate niche-based EC techniques into LES.

In this framework, sampled individuals are projected into an interpretable feature space, where clustering techniques are applied to indirectly partition the language search space. Once partitioned, effective search strategies are designed to allocate sampling resources across niches. The key components of the framework are as follows:

1. **Population initialization**: Generate an initial population of candidate algorithms. This population can be derived from an existing database or dynamically generated during the search process.
2. **Feature space projection**: Project individuals to a feature space using a chosen representation. This step connects the abstract language space with a computationally manageable space.
3. **Niche construction**: Apply clustering methods in the feature space to group individuals based on similarity, thereby forming niches that serve as indirect partitions of the language search space.
4. **Collaborative search**: Perform iterative search using both *prompt-centric operators* and *EC-inspired operators*, continuing until the target problem is solved. These operators enhance the efficiency of algorithm discovery in complementary ways:
   - *Prompt-centric operators* exploit the contextual understanding capabilities of LLMs to guide the sampling process at the language level. By enriching few-shot prompts with additional information (such as reflections on past samples or summaries of search trajectories), these operators steer the LLM toward generating more promising candidate algorithms. Existing LES methods primarily focus on this strategy [7, 8].
   - *EC-inspired operators* emulate established evolutionary mechanisms to maintain diversity and ensure global search capabilities. These operators use the niche structure to select parent individuals and apply evolutionary strategies both within and across niches. This design enables efficient resource allocation and promotes the discovery of diverse, high-quality algorithms. The integration of niche-aware EC operators is a core contribution of this work.

Importantly, the proposed framework is highly flexible and extensible. It allows integration with existing LES methods or mature EC techniques to enhance intra-subregion sampling, refine LLM prompt engineering, and improve resource allocation across subregions. Additionally, any feature

and clustering method can be utilized. In this work, we introduce PartEvo, a specific implementation that demonstrates the effectiveness of search space partitioning in automated algorithm discovery.

## 3.2 PartEvo

We instantiate the general framework with *Partition to Evolve* (PartEvo), an LES method that integrates niche-based collaborative search with advanced prompt design for automated algorithm discovery. Given a problem specification as input, PartEvo outputs high-quality algorithms tailored to the task. The overall architecture of PartEvo is shown in Fig. 1, with pseudocode and a concrete example provided in Appendix C.

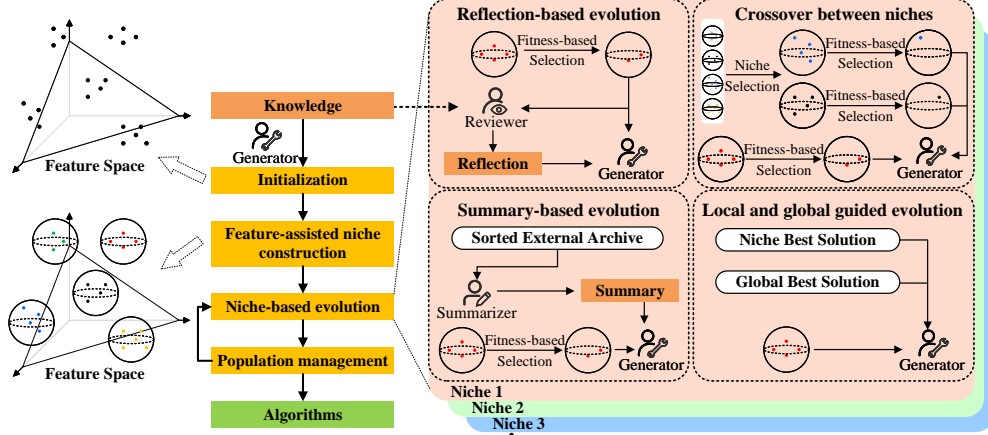

Figure 1: Overview of the PartEvo framework. The left section illustrates the ideal distribution and clustering results of the algorithms within the feature space. The central part highlights PartEvo's main workflow. The right section provides a detailed illustration of PartEvo's evolutionary operators.

**Individual representation.** PartEvo performs algorithm discovery through an evolutionary process, representing each candidate algorithm as an individual. Each individual consists of both code and its corresponding "thoughts", following the design introduced in EoH [7]. This dual-format representation is adopted for two reasons: it leverages the LLM's strong language understanding and broad knowledge base, and it enables feature projection from both the algorithm's code and its natural language rationale. This flexibility is beneficial for subsequent feature-assisted niche construction. Concrete examples of individuals are provided in Appendix H.

**Population initialization.** PartEvo initializes individuals in batches, iteratively constructing the initial population to reach a target size of $N$. During the generation of each batch, the "thoughts" of all previously generated individuals (if any) are embedded into the few-shot prompt to guide the LLM, helping avoid redundant outputs. This approach encourages diversity and promotes broad coverage of the search space. In addition, PartEvo maintains an external archive of individuals, ranked by performance, with a fixed size $E$. This archive stores search trajectories and provides valuable contextual information that helps the LLM develop a more refined understanding of the search space.

**Feature-assisted niche construction** PartEvo employs $K$-means clustering [41] to group initial individuals based on their feature representations, thereby constructing $K$ niches. The number of niches $K$ acts as a control knob, allowing users to adjust the granularity of partitioning. For a fixed population size $N$, different values of $K$ correspond to varying degrees of sampling resource concentration, thus balancing exploration and exploitation in the search space. We explore two feature representations as illustrative examples:

1. Code Similarity Vector: Each individual is represented by a vector capturing its similarity to all others in the population. This captures global relationships between individuals, ensuring that similar individuals are placed close to each other.
2. Thought Embedding: The "thoughts" of individuals are embedded into a vector space using word embeddings, providing a semantic representation of their design rationale.

We adopt $K$-means to present our main experimental results due to its simplicity and transparency, ensuring that observed performance improvements primarily reflect the effects of niche-based evolution rather than complex partitioning. Additional experiments with other clustering methods are provided in **Appendix B.8**. Further details on feature projection are provided in **Appendix E**.

**Evolutionary operators.** As illustrated in the right section of Fig. 1, PartEvo employs two types of operators within each niche—*prompt-centric* and *EC-inspired operators*—detailed as follows:

(I) Reflection-based Evolution (RE): A subset of $m$ individuals is selected, and the *Reviewer* generates reflective feedback for each. These individuals and their corresponding feedback are provided as few-shot prompts to the *Generator*, which produces $m$ new individuals. This operator drives improvement by guiding individuals to reflect on and revise their past designs.

(II) Summary-based Evolution (SE): The *Summarizer* analyzes the external archive to assess the overall algorithmic landscape, identifying both promising and unproductive subregions. The resulting summary, combined with a selected subset of $m$ individuals, is used to construct few-shot prompts that guide the *Generator* in producing $m$ new individuals. This operator facilitates informed exploration by incorporating global knowledge of the search space. It draws inspiration from classical ideas such as information sharing and tabu search [42].

(III) Crossover between Niches (CN): $k$ niches are randomly selected, and one representative individual is drawn from each. These $k$ individuals are paired with $m$ individuals from the current niche to form $m$ pairs. Each pair then prompts the *Generator* to produce a new individual, promoting cross-niche knowledge transfer and enhancing population diversity.

(IV) Local and Global guided Evolution (LGE): Each individual is paired with both the best-performing individual in its niche and the global best in the population. These triplets are then formatted as few-shot prompts to generate new individuals. This design ensures that all individuals benefit from guidance toward high-performing regions, thus promoting faster convergence.

Operators RE and SE are classified as *prompt-centric*: they employ auxiliary LLMs to provide "verbal gradients" that steer generation in semantically meaningful directions. In contrast, CN and LGE are *EC-inspired*: they incorporate the principles of niche collaboration and evolutionary heuristics to improve search efficiency. Detailed prompts for these operators are provided in **Appendix D**.

**Resource allocation strategy** In PartEvo, sampling resources (e.g., queries to LLMs) are distributed evenly across niches. Within each niche, a finer-grained resource allocation is implemented through elite-preserving probabilistic selection. The best individual in each niche is always chosen as a parent, while the remaining parents (when $m>1$) are selected probabilistically based on their performance. This strategy not only encourages the exploitation of high-performing candidates within each niche but also ensures the exploration of less-explored subregions. Unlike greedy parent selection, which concentrates resources solely on the best-performing individuals, PartEvo's niche-based evolution allows dynamic control over resource allocation, effectively balancing exploration and exploitation. This is also the rationale behind the name *Partition to Evolve*.

**Population management.** In each iteration, the $K$ niches independently execute the four operators, collectively generating $N+3*m*K$ new individuals. Each niche selects individuals using a no-replacement roulette wheel mechanism, ensuring the total population size remains constant.

## 4 Experiments

This section presents PartEvo's performance on algorithm discovery tasks. Additionally, we conduct thorough ablation studies to assess the contributions of niching, feature-based niche construction, and the proposed operators. The results demonstrate that PartEvo, which integrates advanced prompting strategies with niche-based EC techniques, achieves significant improvements in search efficiency.

### 4.1 Experimental settings

**Benchmarks.** We assess PartEvo's ability to design meta-heuristic algorithms [43] using four benchmarks. In each case, the LES method is tasked with generating algorithms to solve the given problem. With the same sample budget, the effectiveness of the LES method is proportional to the

quality of the solutions produced by the algorithms it designs. A brief overview of each benchmark is provided below, with detailed descriptions in Appendix G.

(I) Unimodal optimization problems (P1): This benchmark assesses the LES method's ability to design algorithms for problems with a single global optimum and no local optima. It evaluates the efficiency of the generated search strategies in simple optimization landscapes.

(II) Multimodal optimization problems (P2): This benchmark focuses on problems with multiple local optima, aiming to evaluate whether the LES-designed algorithms can avoid premature convergence and maintain sufficient exploration of the search space.

(III) Task offloading in mobile edge computing systems (P3): This real-world benchmark involves solving task offloading problems under constraints such as execution time, energy consumption, and bandwidth [44]. It evaluates the LES method's ability to generate algorithms for mixed-integer nonlinear programming (MINLP) problems with multiple real-world constraints.

(IV) Machine-level scheduling for heterogeneous plants (P4): This problem requires designing algorithms to handle scheduling tasks across machines with different capabilities and managing transportation of products [45]. It tests the LES method's ability to design algorithms that solve high-dimensional nonlinear integer programming problems subject to complex industrial constraints.

**Baselines.** We compare PartEvo against three LES methods and three human-designed meta-heuristic algorithms. The peer LES methods include Funsearch, EoH, and ReEvo.

- Funsearch [6] uses an island-based evolutionary approach to iteratively improve function quality, with each island acting as a niche formed through random grouping. Our comparison with Funsearch highlights that feature-assisted niche construction effectively balances exploration and exploitation.

- EoH [7] models algorithm discovery as a search problem in the dual space of algorithms, represented both as code and as thought. Direct comparison with EoH demonstrates the superior search efficiency of PartEvo, which is attributed to the integration of advanced prompt design and niche-based EC techniques.

- ReEvo [8] introduces "verbal gradients" into the LES paradigm via genetic cues in natural language form, enabling more focused searches within the language-based search space. A direct comparison with ReEvo highlights the superior search efficiency achieved through the integration of advanced EC techniques.

The human-designed meta-heuristic baselines include enhanced variants based on Genetic Algorithm, Differential Evolution, and Particle Swarm Optimization [46–48]. For simplicity, we refer to them as GA, DE, and PSO in the experimental settings, with all variants tuned to provide strong human-designed baselines. A more detailed description of these algorithms is provided in **Appendix F**. By comparing PartEvo against human-designed baselines, we provide compelling evidence of its effectiveness in designing algorithms for both synthetic and real-world optimization problems. Such comparisons are essential to assess whether LES methods can surpass established human expertise and offer practical value as automated design tools.

**Implementation details.** We frame all benchmarks as black-box optimization tasks and provide them to LES methods. Each LES method receives only performance feedback for the algorithms it generates, without access to the mathematical models or instance-specific details. This prevents the LLM from exploiting explicit knowledge of the problem to overfit or cheat. Each benchmark includes multiple instances, split into training and testing sets. LES methods develop algorithms on the training instances, and their performance is evaluated on the testing set. This setup allows us to assess whether the LES methods overfit training instances, thereby mitigating potential biases arising from fixed-pattern solutions that could inflate performance.

For all LES methods, the number of candidate algorithms sampled by LLMs is set to 500. Each benchmark is independently solved over four runs to reduce the effect of randomness. All LES methods utilize the pre-trained `GPT-4o-mini` model. The resulting meta-heuristics are constrained to 30,000 candidate solution evaluations and a maximum runtime of 180 seconds. EoH and PartEvo both use a population size of 16. Funsearch, EoH, and ReEvo follow their respective default configurations provided by the LLM4AD platform [49]. Unless otherwise explicitly stated, all results reported for PartEvo in this paper utilize *Code Similarity* as the feature for its niche construction process.

## 4.2 Comparative evaluations

We evaluate the ability of PartEvo, Funsearch, EoH, and ReEvo to design meta-heuristic algorithms for identical instances. The generated algorithms are also compared against human-designed baselines. Table 1 reports the best results across four runs on training and testing sets. Lower objective values indicate better performance, reflecting how efficiently each designed meta-heuristic utilizes the fixed budget of 30,000 evaluations. For reference, the *Optimal* column lists the known optima, representing the minimum achievable objective values for each benchmark. Additional statistics, including mean and standard error of the mean across runs, are provided in Table 2 and illustrated in Fig. 2.

Table 1: Best results from multiple runs on training and testing instances for each benchmark

| Benchmark | Instances | GA | PSO | DE | Funsearch | EoH | ReEvo | PartEvo | *Optimal* |
|---|---|---|---|---|---|---|---|---|---|
| P1 | Training | 20.293 | 15.569 | 0.052 | 0.418 | **0.000** | **0.000** | **0.000** | 0.000 |
| | Testing | $5.686 \times 10^6$ | **0.000** | **0.000** | 0.604 | **0.000** | 2.253 | **0.000** | 0.000 |
| P2 | Training | 1772.61 | 890.99 | 881.44 | 800.25 | **800.00** | 801.32 | **800.00** | 800.00 |
| | Testing | 585.58 | 421.76 | 386.50 | 376.36 | 462.08 | 375.57 | **344.62** | 310.00 |
| P3 | Training | 31447.05 | 8805.09 | 6541.56 | 10503.67 | 7003.53 | 6589.08 | **6471.31** | - |
| | Testing | 75279.85 | 64414.61 | 58704.47 | 60892.32 | 58327.20 | 59061.28 | **56876.40** | - |
| P4 | Training | $5.29 \times 10^6$ | 30980.3 | 25088.2 | 20425.0 | 3697.6 | 10630.3 | **2792.1** | - |
| | Testing | $7.31 \times 10^7$ | $2.41 \times 10^5$ | $2.37 \times 10^5$ | $1.49 \times 10^5$ | 34891.9 | $1.34 \times 10^5$ | **14396.1** | - |

As shown in Table 1, PartEvo consistently outperforms both peer LES methods and human-designed baselines across all benchmarks. On complex problems such as P3 and P4, it demonstrates a clear advantage over Funsearch, EoH, and ReEvo. Notably, on P3, it is the only LES method that surpasses the DE baseline. On the most challenging task, P4, it achieves a 90.1% reduction in manufacturing cost compared to the best-performing DE variant.

Table 2: Average results and standard error of the mean from multiple runs on training instances

| Benchmark | Funsearch | EoH | ReEvo | PartEvo |
|---|---|---|---|---|
| P1 | 6.30($\pm$2.11) | 1.43($\pm$1.40) | 1.64($\pm$1.13) | **0.00($\pm$0.00)** |
| P2 | 810.88($\pm$5.26) | 800.36($\pm$0.33) | 804.52($\pm$1.60) | **800.01($\pm$0.01)** |
| P3 | 10080.6($\pm$226.40) | 7301.9($\pm$251.60) | 7074.5($\pm$392.56) | **6583.8($\pm$79.74)** |
| P4 | 30094.6($\pm$3485.37) | 17679.9($\pm$3082.57) | 15345.5($\pm$2381.9) | **5171.6($\pm$2189.14)** |

Table 2 presents the mean and standard error of the mean (SEM) of results across multiple runs on training instances, complementing the data shown in Table 1. PartEvo consistently delivers the best average performance with relatively small SEM values, indicating strong algorithm design and stability. Overall, PartEvo outperforms peer methods across all four benchmarks.

Fig. 2 illustrates the convergence behavior of the best objective values for PartEvo, Funsearch, EoH, and ReEvo across four runs, showing their performance over 500 samples, which reflects the algorithm design efficiency. Colored curves show mean performance across runs, and shaded areas denote variance. Dashed horizontal lines mark human-designed baselines. PartEvo not only reaches superior solutions with fewer samples but also maintains the smallest variance, indicating strong robustness and consistency. Compared to Funsearch, which also uses subpopulation, PartEvo continues to improve throughout the search, suggesting feature-assisted niche construction more effectively avoids local optima. When compared to ReEvo, which uses the concept of verbal gradient but relies on a basic EC framework, PartEvo shows a higher search efficiency. This highlights that advanced EC techniques can significantly enhance search performance and should be given equal emphasis as prompt engineering in the design of the LES paradigm.

In summary, PartEvo achieves the fastest convergence, the lowest final objective values, and the highest stability across runs. These results demonstrate the effectiveness of combining niche-based evolutionary search with LLM guidance for domain-specific algorithm discovery. To control for the influence of initialization, additional experiments using identical initial populations are conducted (see **Appendix B.2**). Further results on additional benchmarks are provided in **Appendix B.3**. Comparative evaluations of these LES methods with a 2000-sample budget are presented in **Appendix B.6**.

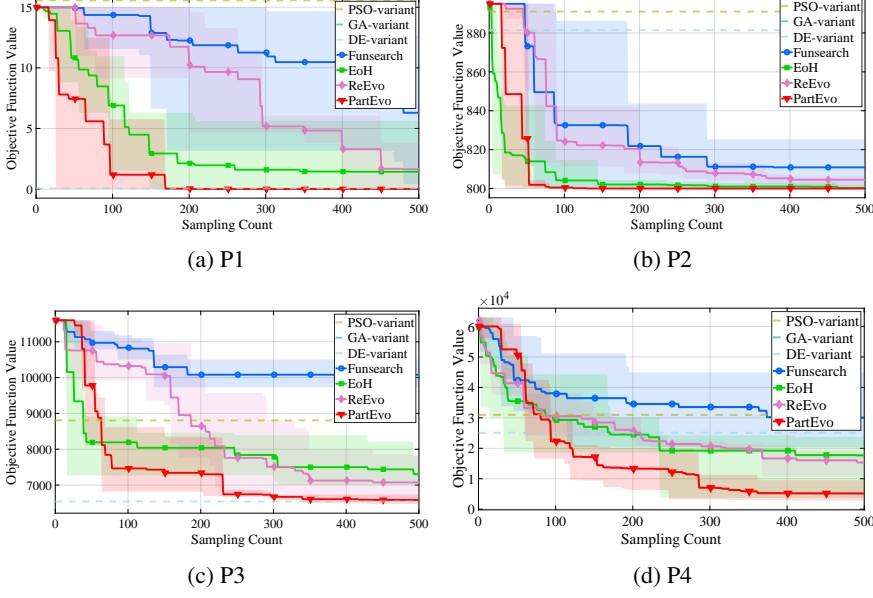

Figure 2: Convergence of LES methods across four runs on benchmarks (P1–P4).

## 4.3 Sampling similarity analysis

We assess PartEvo's ability to balance exploration and exploitation by analyzing the similarity of its sampled results. Table 3 reports the similarity scores for PartEvo, Funsearch, EoH, and ReEvo, measured over the first and last 50 samples out of 500 total sampling steps. The score is measured using CodeBLEU [50], a metric that quantifies the similarity of abstract syntax trees among sampled individuals. The results reveal that each LES method exhibits distinctive and consistent sampling patterns across different problem settings.

Table 3: Similarity scores of sampling results for PartEvo, Funsearch, EoH, and ReEvo

| Benchmark | First 50 samples | | | | Last 50 samples | | | |
|---|---|---|---|---|---|---|---|---|
| | Funsearch | EoH | ReEvo | PartEvo | Funsearch | EoH | ReEvo | PartEvo |
| P1 | 0.64 | 0.54 | 0.65 | **0.36** | 0.57 | 0.58 | 0.71 | 0.70 |
| P2 | 0.60 | 0.55 | 0.65 | **0.38** | 0.56 | 0.60 | 0.71 | 0.67 |
| P3 | 0.62 | 0.54 | 0.66 | **0.41** | 0.57 | 0.58 | 0.69 | 0.60 |
| P4 | 0.61 | 0.54 | 0.63 | **0.41** | 0.58 | 0.60 | 0.60 | 0.60 |

PartEvo exhibits substantially lower similarity scores in the early stage compared to the other LES methods, indicating broader coverage of diverse subregions at the beginning of the search. As the search proceeds, PartEvo's similarity scores rise, reflecting a natural shift toward convergence, where niches begin to share more effective and practical techniques. This dynamic adjustment highlights PartEvo's ability to transition from wide exploration to focused exploitation.

By contrast, Funsearch's island-based evolution encourages exploration across multiple subpopulations, but the lack of feature-assisted niche construction results in less targeted resource allocation. EoH, which relies solely on fitness-based parent selection, maintains consistently high similarity scores throughout the process, suggesting a bias toward exploitation with limited exploration. ReEvo, by excessively adhering to accumulated long- and short-term experience, also produces persistently high similarity, leading to reduced diversity.

The superior performance of PartEvo, as discussed in **Subsection 4.2**, underscores the benefits of its batch initialization strategy and explicit feature-assisted niche construction. By effectively balancing exploration and exploitation, PartEvo aligns with the ideal search paradigm: wide exploration in the early stages followed by concentrated exploitation as the search converges.

## 4.4 Ablation study

We conduct a comprehensive ablation study to assess the contributions of key components in PartEvo.

**Effectiveness of Niching.** Table 4 reports the performance of PartEvo when dividing a population of size 16 into 1, 2, 4, and 6 niches. The number of niches, denoted as $K$, controls the degree of resource distribution during the sampling process. Specifically, $K=1$ indicates no partitioning, where the entire population is treated as a single niche. When $K$ is small, resources are concentrated on individuals with lower objective values, enabling more intensive exploitation of specific regions of the search space. Conversely, when $K$ is large, resources are more dispersed across the search space. Such over-dispersion reduces the sampling intensity in promising regions, leading to diminished search efficiency due to excessive exploration. The results in Table 4 empirically confirm this trade-off.

Table 4: Effect of niching granularity on PartEvo performance

| Benchmark | $K=1$ | $K=2$ | $K=4$ | $K=6$ |
|-----------|-------|-------|-------|-------|
| P1 | 9.379 | 4.544 | **0.000** | 8.241 |
| P2 | 800.00 | 800.01 | **800.00** | 800.11 |
| P3 | 6474.07 | 6549.09 | **6471.31** | 6553.08 |
| P4 | 13418.7 | 10572.0 | **4539.1** | 12427.2 |

For a population of size 16, setting $K=4$ achieves the best balance between exploration and exploitation, significantly improving the algorithm discovery process. When $K=1$, PartEvo reduces to a strategy that selects parents solely based on fitness values, similar to EoH. While this approach works well for simple problems (e.g., P2), where iterative refinements on any baseline algorithm can easily reach the optimal solution, it struggles with more complex problems. In such cases, the lack of exploration causes the search to stagnate in local optima, limiting the discovery of high-quality algorithms. When $K=6$, the excessive dispersion of sampling resources results in performance degradation, as promising subspaces fail to receive sufficient attention. These findings emphasize the importance of niching in improving search efficiency. However, the granularity of niching must be carefully tuned to strike an optimal balance between exploration and exploitation. Based on our experiments, we find that for a population size of 16, $K=4$ achieves the best performance.

Table 5: Effect of feature usage on PartEvo performance

| Benchmark | Random | Code Similarity | Thought Embedding |
|-----------|--------|-----------------|-------------------|
| P1 | 3.974 | 0.074 | **0.000** |
| P2 | 803.92 | 800.42 | **800.00** |
| P3 | 6607.99 | **6539.24** | 6598.15 |
| P4 | 17872.1 | **6424.7** | 13213.8 |

**Importance of feature-assisted niche construction.** Table 5 compares the performance of PartEvo using feature-assisted partitioning versus random partitioning for niche construction. Results are averaged over three independent runs, each initialized with the same population. The best performance for each benchmark is highlighted in bold, with the second-best underlined. The results demonstrate that PartEvo consistently achieves the best or second-best performance across all benchmarks when using *Code Similarity* and *Thought Embedding*. This suggests that these features effectively capture the structural properties of the language space, leading to more informed partitioning. In contrast, random partitioning, which lacks such structural guidance, performs significantly worse. These findings emphasize the importance of meaningful features in informing partitioning strategies. A more detailed analysis of the convergence behavior under different features is provided in **Appendix B.4**.

**Effectiveness of prompt-centric and EC-inspired operators.** We conducted an ablation study by systematically disabling specific operators to evaluate the effectiveness of prompt-centric and EC-inspired operators. Table 6 reports the performance degradation ($\Delta$Perf) relative to the full PartEvo. PartEvo♠ serves as a baseline version, retaining only crossover and mutation operations, excluding advanced prompt and niche-based strategies. The results clearly indicate that disabling advanced

prompt engineering or niche-based collaborative search leads to significant performance drops, with the most severe degradation occurring when both are removed simultaneously. These findings highlight the crucial role of integrating EC-based techniques into the language space to enhance PartEvo's performance. Detailed ablation results for each operator are provided in **Appendix B.5**.

Table 6: Performance degradation in ablation study of PartEvo

| Method | P1 | P2 | P3 | P4 | $\Delta$Perf (%) |
|---|---|---|---|---|---|
| Full PartEvo | **0.074** | **800.00** | **6539.24** | **6424.7** | 0.00 |
| w/o prompt-centric operators | 1.585 | **800.00** | 6946.69 | 15259.2 | 553.74 |
| w/o EC-inspired operators | 1.288 | 807.37 | 7651.28 | 14324.7 | 451.31 |
| PartEvo♠ | 7.248 | 807.58 | 8524.38 | 17924.0 | 2509.76 |

## 5 Discussions and limitations

**Dependency on large language models** The performance of PartEvo is inherently tied to the capabilities of the underlying LLMs. Specifically, prompt-centric operators, such as RE (Reflection) and SE (Summarization), depend on the LLM's ability to perform sophisticated contextual reasoning and leverage expert knowledge in algorithm design. As LLMs continue to evolve, we anticipate improvements in PartEvo's algorithm discovery capabilities, a trend also evidenced by comparative results across different LLMs (**Appendix B.7**). However, it is important to highlight that the integration of EC-inspired operators, such as CN and LGE, helps mitigate some of this dependency. These operators enable more refined parent selection processes and use compact prompts that capitalize on the LLM's core strengths [51], reducing reliance on complex summarization abilities.

**Understanding the language search space** PartEvo indirectly analyzes and partitions the language search space through feature-assisted niche construction. It treats algorithm discovery as a black-box optimization task, iteratively sampling candidate algorithms within the language space using an evolutionary process. A further pursuit is to enhance the understanding of the structure and landscape of this language space. Achieving this requires more sophisticated feature mapping techniques to better analyze and partition the space (PartEvo currently utilizes code similarity and thought embeddings as examples). It is worth noting that PartEvo can serve as a valuable platform for validating novel language space mapping techniques. New algorithm similarity or landscape analysis methods can be easily integrated into the framework. As discussed in **Appendix B.4**, improvements in niche partitioning (i.e., the ability to partition the language search space effectively) will likely lead to more stable and improved performance on specific algorithm design tasks.

Additional discussions and extended analyses are provided in **Appendix A.2**.

## 6 Conclusion

This work introduces a general Large Language Model-assisted Evolutionary Search (LES) framework with abstract search space partitioning, enabling LES methods to effectively integrate niche-based Evolutionary Computation (EC) techniques within language spaces. Building on this framework, we present PartEvo, which combines advanced prompting strategies with both local and global search methods. Applying PartEvo to four optimization benchmarks demonstrates that it yields competitive meta-heuristic algorithms that outperform human-designed ones. Furthermore, it surpasses prior LES methods, particularly in resource-constrained settings. Crucially, this work, using niching as a concrete example, proves the feasibility and significant benefits of integrating mature EC techniques into the LES paradigm. Future work will focus on refining abstract partitioning techniques and developing more specialized EC search strategies to further enhance the LES method's capabilities in automating algorithm discovery. The source code can be found in https://github.com/QingL2000/PartEvo.

## Acknowledgments and Disclosure of Funding

This work was supported by the General Research Fund (GRF) of the Research Grants Council (RGC) of Hong Kong (Project No. CityU 11217325).

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

# A    Discussion: Comparison, Limitation, and Impact Statement

## A.1    Comparisons with Funsearch, EoH, and ReEvo

Funsearch [6] is one of the earliest LLM-assisted Evolutionary Search (LES) methods. Prior to this, the Algorithm Evolution using Large Language Model (AEL) [32] explored the use of prompt engineering to guide LLMs as evolutionary operators for searching knowledge-rich language spaces.

EoH [7] keenly recognized that LLMs can assist in searching the code space through higher-level thoughts. Consequently, EoH represents individuals using both code and thoughts, exploring the coevolution of thoughts and code within an evolutionary framework, which leads to superior performance while mitigating computational costs.

ReEvo [8] demonstrated that LLMs can provide verbal gradients through genetic cues, enabling more directed searches within the language search space. This approach effectively implements gradient-based search techniques in the language domain.

In terms of search strategies, Funsearch employs an island-based evolutionary approach to iteratively improve function quality by generating multiple islands in a random manner [6]. This island management promotes exploration and reduces the risk of premature convergence. In contrast, both EoH and ReEvo follow a single-population evolutionary framework, focusing on leveraging the capabilities of LLMs.

Our work introduces niche-based collaborative search mechanisms from evolutionary computation into LES, further enhancing search efficiency within language spaces. The proposed framework is general and can be easily integrated with existing LES approaches, with PartEvo serving as a specific instantiation of these core ideas. PartEvo first introduces feature-assisted niche construction to indirectly partition the language space into multiple subregions. It then defines prompt-centric operators (RE and SE) and EC-inspired operators (CN and LGE) for algorithm discovery processes. In the prompt-centric operators, LLMs are guided to generate promising algorithms through verbal gradients. The EC-inspired operators incorporate principles of niche collaboration and evolutionary heuristics to improve search efficiency. Experimental results demonstrate that the integration of niche-based techniques significantly improves algorithm discovery efficiency by over twofold, making LES more practical.

## A.2    Additional discussion and limitation

**Scenarios where PartEvo may not be suitable**    A key requirement for PartEvo is the continuous validation of the generated algorithms. This requires that the problems to be solved are formulated as mathematical models or situated within environments that can be simulated. For example, with a sample budget of 500 and an evaluation budget of 30,000, a total of 15,000,000 evaluations would be required to assess the objective value of the solutions. For some expensive optimization problems (e.g., parameter optimization in computational fluid dynamics [52] or drug design [53]), the extensive evaluations required by PartEvo could result in prohibitive computational costs. A promising direction to address this is the use of surrogate models [54] to approximate objective values, which could accelerate the evaluation process and alleviate the computational burden. Alternatively, designing algorithms for similar, more computationally feasible problems and then transferring these algorithms to expensive optimization tasks may also provide an effective strategy. For large-scale problems, employing a divide-and-conquer approach [55] at a higher level to reduce problem complexity could enable PartEvo to focus on solving lower-level subproblems more efficiently.

**Cross-paradigm discussion**    The LES is an emerging paradigm for automated algorithm discovery (AAD). This work primarily focuses on addressing the challenges of incorporating advanced evolutionary computation (EC) techniques into the LES paradigm, which is complicated by its abstract, language-based search space. Using the integration of niching techniques as a case study, we demonstrate significant improvements in the performance, efficiency, and robustness of the LES paradigm for AAD when successfully applying these advanced EC methods. However, the field of AAD also includes other established paradigms, such as Genetic Programming (GP) [56], Deep Learning (DL) [3], and Reinforcement Learning (RL) [4], which merit a more systematic comparison with the LES paradigm. Incorporating insights from related works [7, 28, 57–59], we provide a qualitative, literature-informed comparison across several key aspects in Table 7, highlighting these

distinctions. A comprehensive, systematic cross-paradigm comparative study is a valuable direction for future research, and we are committed to pursuing it. Existing work has already begun to explore such comparisons [28], and we plan to conduct a more thorough analysis to better understand the trade-offs between these distinct paradigms.

Table 7: A qualitative comparison of AAD paradigms

| Feature | LES | GP | DL or RL |
|---------|-----|----|---------| 
| Manual Design Cost | **Low.** The framework automatically generates the algorithms based on a high-level task description, requiring minimal human effort. | **High.** Requires significant human effort to define problem-specific operators, functions, and grammar for the algorithm search space. | **High.** Requires extensive design of network architectures, training pipelines, and domain-specific datasets. |
| Algorithm Design Efficiency | **High.** A new task can be solved in hours by relying on iterative LLM inference, without the need for model training or fine-tuning. | **High.** The core process is based solely on evolutionary operations, which are computationally fast on a per-step basis, but can still require many generations to find a good algorithm. | **Low.** Requires extensive training periods, often spanning days or weeks, for each specific task. |
| Resource requirements | **High** (requires LLM inference). However, LLM inference tasks can be offloaded to third-party providers via API calls, reducing local hardware requirements. | **Low.** Runs on standard CPUs or GPUs, often without needing high-end hardware. | **High.** Requires substantial GPU resources and large-scale parallel computation, especially during the training phase. |
| Extensibility | **High.** Leverages the vast knowledge embedded in LLMs, allowing strong generalization to diverse algorithm design tasks with minimal adaptation. | **Low.** Generalizing to a new problem often requires extensive manual redesign of the grammar and operators to build the algorithm space for the target problem. | **Low.** Generalizing to a new problem necessitates substantial data collection and lengthy retraining periods. |

## A.3 Extended applications

PartEvo excels at searching for feasible solutions within the language space. In this work, we utilize PartEvo to design meta-heuristic algorithms from the ground up. In fact, any design task at the language level can be executed using this framework, including planning [60], strategy formulation [61], algorithm refinement, and more. Moreover, if LLMs are extended to those with multimodal capabilities [62], feature-assisted niche construction can still be applied within image spaces. In this case, tasks such as industrial design and blueprint drafting could also be automated.

## A.4 Impact statement

This paper advances the field of automated algorithm discovery by integrating EC-based evolutionary techniques with Large Language Models. The proposed method enhances search efficiency within language search spaces, offering significant potential for cost reduction and practical applications across diverse industries. Moreover, the algorithm evolution process generates valuable data, which can further accelerate research and development in this domain. By automating aspects of algorithm discovery, the role of human experts may transition from adapting algorithms for specific applications to focusing on higher-level innovation. However, this shift raises important considerations, including potential changes in labor demands and ethical challenges related to transparency and fairness in

AI-generated systems. We advocate for continued research to address these concerns and to promote the responsible and equitable deployment of this technology.

## B  Additional experiment results

### B.1  Hyperparameters for PartEvo

Unless otherwise stated, we adopt the parameters in Table 8 for PartEvo runs. The entire experiment, including the implementation of PartEvo, EoH, and Funsearch for all four benchmarks, was developed in Python and executed on a single CPU (Intel i9-13980HX) with 32GB of RAM.

Table 8: Hyperparameter settings for PartEvo.

| Hyperparameter | Parameter Description | Value |
|---|---|---|
| $K$ | Number of niches to partition the population into | 4 |
| $k$ | Number of cooperating niches selected in the CN operator | 2 |
| $m$ | Number of parents selected in each evolution step | 2 |
| $N$ | Population size | 16 |
| $E$ | Size of external archive | 40 |
| LLM | Version of LLM used in the evolutionary operators | GPT-4o-mini |
| LLM temperature | Hyperparameter controlling the randomness of LLM text generation | 1 |

### B.2  Comparative evaluations under identical initialization

To ensure a rigorous comparison of the search behaviors of PartEvo, Funsearch, and EoH, we evaluate their performance using the same initial population. The initial population is iteratively generated by PartEvo and then shared across all other LES methods. Each method is independently run three times with the same initial population. The average results of these independent runs are summarized in Table 9. Even when eliminating the randomness introduced by initialization, PartEvo consistently outperforms the other methods across all cases. It is noteworthy that the performance of Funsearch, EoH, and ReEvo shows significant improvement when utilizing the initial population generated by PartEvo. This finding highlights the effectiveness of the batch initialization strategy employed by PartEvo.

Table 9: Average results from multiple runs on training and testing instances for each optimization problem (Under identical initialization)

| Problem | Instances | Funsearch | EoH | ReEvo | PartEvo |
|---|---|---|---|---|---|
| P1 | Training | 9.769 | 1.846 | 1.572 | **0.074** |
|    | Testing | 0.879 | 1.646 | 1.013 | **0.023** |
| P2 | Training | 800.09 | 800.98 | 800.00 | **800.00** |
|    | Testing | 439.31 | 368.95 | **364.19** | 365.20 |
| P3 | Training | 7042.21 | 7268.22 | 6564.44 | **6539.24** |
|    | Testing | 57696.53 | 61142.93 | 57455.34 | **56753.77** |
| P4 | Training | 24289.3 | 19152.4 | 11102.3 | **6424.7** |
|    | Testing | $1.67 \times 10^5$ | 77134.3 | 79795.2 | **62731.4** |

Figure 3 presents the convergence of the best objective function values over 500 samples for PartEvo, Funsearch, and EoH. Under the same initial population, the convergence behavior mirrors that observed in **Subsection 4.2**. Notably, PartEvo achieves superior results with fewer samples.

### B.3  Comparative evaluations on Online Bin Packing Benchmark

In the main paper, we established four benchmarks (P1-4) to showcase PartEvo's design capabilities for meta-heuristic algorithms. To provide a more comprehensive evaluation, we further assess PartEvo

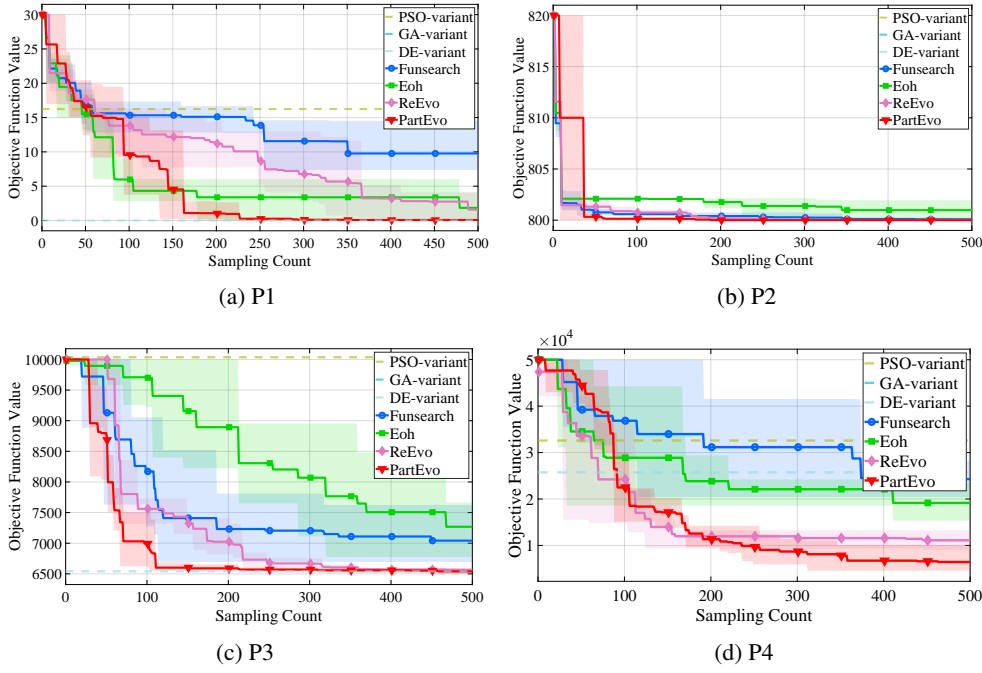

(a) P1

(b) P2

(c) P3

(d) P4

Figure 3: Convergence behavior of different LES methods using the same seed across four independent runs on Problems P1, P2, P3, and P4

alongside peer LES methods on the Online Bin Packing Benchmark for heuristic design, as utilized in EoH and Funsearch studies. Additionally, we compare the recent ReEvo approach on this task.

For consistency, a sample budget of 500 is applied across all methods, with a population size of 16, conducting four independent runs for each algorithm. Table 10 presents the fraction of excess bins (as a percentage) relative to the lower bound (with lower values indicating better performance) for various bin packing heuristics applied to Weibull instances.

Table 10: Average results from multiple runs on the Bin Package Benchmark (Lower is Better)

| Method | 1k_C100 | 1k_C500 | 5k_C100 | 5k_C500 | 10k_C100 | 10k_C500 |
|---|---|---|---|---|---|---|
| First fit | 5.32 | 4.97 | 4.40 | 4.27 | 4.44 | 4.28 |
| Best fit | 4.87 | 4.50 | 4.08 | 3.91 | 4.09 | 3.95 |
| EoH | 4.76 | 4.38 | 2.32 | 2.18 | 2.04 | 1.97 |
| Funsearch | 4.07 | 4.69 | 3.19 | 3.56 | 3.16 | 3.44 |
| ReEvo | 4.24 | 3.94 | 2.13 | 2.07 | 2.07 | 1.96 |
| PartEvo | **3.51** | **3.23** | **1.14** | **1.11** | **0.84** | **0.79** |

First Fit and Best Fit are two human-designed heuristics. All LES methods surpass these baselines. The performance of all LES methods on the Bin Package Benchmark is consistent with the results from the four benchmarks presented in the main paper. PartEvo achieves the best average performance, proving its high efficiency in solving AAD tasks under limited budget constraints.

Notably, PartEvo and ReEvo outperform EoH and Funsearch, underscoring the advantages of Reflection and Summary. However, due to its unique EC-inspired operators (CN and LGE), PartEvo surpasses ReEvo. This highlights that both "verbal gradient" and advanced EC techniques can enhance search efficiency and should be equally emphasized.

## B.4 Detailed feature analysis

Figure 4 presents the convergence behavior of the best objective values achieved by PartEvo when utilizing different feature sets on problems P1 and P2. Each feature set is evaluated through three

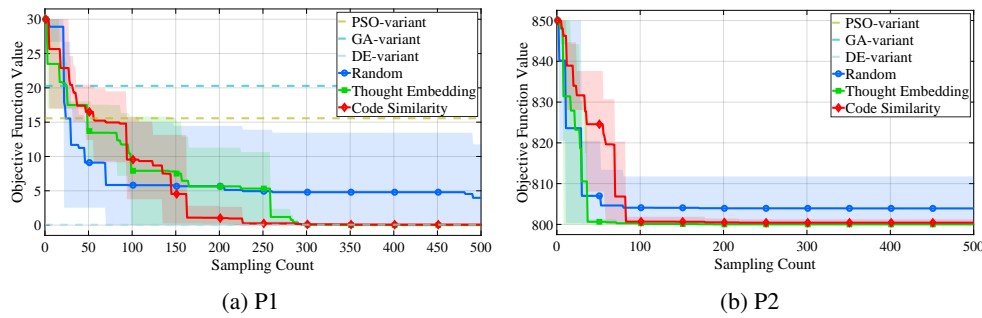

| (a) P1 | (b) P2 |
|---|---|

Figure 4: Impact of partitioning features on convergence in PartEvo on P1 and P2

independent runs, all initialized with the same population. As observed, PartEvo, when augmented with the *Code Similarity* and *Thought Embedding* features, consistently converges to high-quality algorithms. In contrast, the use of random features (depicted by the blue curves and shaded regions) leads to a broader range between the upper and lower envelopes, indicating greater instability in algorithm design.

These results emphasize the critical role of feature-assisted niche construction in enhancing the efficiency of automated algorithm discovery. Even with random partitioning, there is a chance that the resulting niches may coincidentally align in a way that facilitates the discovery of high-performing algorithms. This suggests that certain niche structures inherently lead to better outcomes. Notably, Code Similarity and Thought Embedding, as employed in this work, are two viable feature mapping strategies that markedly increase the reliability and efficiency of generating powerful algorithms with PartEvo.

## B.5 Detailed ablation study

To better understand the contributions of the four evolutionary operators in PartEvo, we conducted a comprehensive ablation study. The average performance across multiple runs for each ablation setting is summarized in Table 11. Variants of PartEvo with specific operators removed are denoted using the prefix "w/o" (e.g., w/o RE indicates the removal of the RE operator). For clarity, we restate the definitions of the key variants here, consistent with Subsection 4.4 in the main text:

- PartEvo†: Removes the RE and SE operators, which utilize additional LLMs to enhance prompts through reflection and summarization.
- PartEvo‡: Removes the CN and LGE operators, which implement subspace-based collaborative search.
- PartEvo♠: A baseline version that retains only crossover and mutation operations, excluding advanced prompt and subspace-based search strategies.

Table 11: Ablation study results on four benchmark problems.

| Method | P1 | P2 | P3 | P4 | $\Delta$Perf (%) |
|---|---|---|---|---|---|
| Full PartEvo | **0.074** | **800.00** | **6539.24** | **6424.7** | 0.00 |
| w/o RE | 0.389 | 800.00 | 6915.27 | 8433.5 | 117.74 |
| w/o SE | 1.493 | 800.01 | 6610.28 | 13761.8 | 515.12 |
| PartEvo† | 1.585 | 800.00 | 6946.69 | 15259.2 | 553.74 |
| w/o CN | 0.723 | 800.00 | 6547.89 | 11357.3 | 241.82 |
| w/o LGE | 0.873 | 800.04 | 6849.81 | 10491.9 | 290.98 |
| PartEvo‡ | 1.288 | 807.37 | 7651.28 | 14324.7 | 451.31 |
| PartEvo♠ | 7.248 | 807.58 | 8524.38 | 17924.0 | 2509.76 |

The metric ΔPerf(%) quantifies the performance degradation relative to the full PartEvo, with larger values indicating greater declines. As shown in Table 11, each operator is critical to the overall efficiency of PartEvo.

- Removing either the reflection or summarization enhancement operators (RE or SE) results in significant performance drops (ΔPerf=117.74% and 515.12%, respectively). The impact is even more pronounced when both are removed (ΔPerf=553.74%).
- Similarly, removing the subspace-based collaborative search operators (CN or LGE) leads to substantial degradations (ΔPerf=241.82% and 290.98%, respectively). The effect worsens when both are excluded (ΔPerf=451.31%).
- The baseline version (PartEvo♠), which excludes all advanced operators and relies solely on basic evolutionary operations, experiences the most severe performance degradation (ΔPerf= 2509.76%).

These results underscore the importance of both prompt-centric and EC-inspired operators. By integrating these two aspects, PartEvo effectively combines the strengths of LLMs and niche-based EC techniques, demonstrating its potential as a robust LES method for automated algorithm discovery.

### B.6 Comparative evaluations with an expanded sample budget

PartEvo demonstrates superior efficiency in algorithm discovery under a limited sample budget of 500 LLM queries. To further assess its potential, we expand the budget to 2000 queries and evaluate performance on benchmarks P3 and P4. Benchmarks P1 and P2 are excluded from this analysis, as prior results indicate that 500 queries already suffice for PartEvo, ReEvo, and EoH to design meta-heuristics that approach optimal solutions. All four LES methods are initialized with the same random seed for fairness, as detailed in **Appendix B.2**. This experiment aims to: (i) examine whether PartEvo's early-stage efficiency induces premature convergence that limits long-term performance; and (ii) investigate whether Funsearch, EoH, and ReEvo can exploit the larger budget to substantially improve their outcomes.

Table 12: Average results from three runs on training instances for benchmark P3 and P4

| Benchmark | Budget | Funsearch | EoH | ReEvo | ParEvo |
|---|---|---|---|---|---|
| P3 | 500* | 7042.20 | 7268.21 | 6564.44 | **6539.23** |
|    | 2000 | 6656.7 | 6749.5 | 6506.51 | **6479.49** |
| P4 | 500* | 24289.3 | 19152.4 | 11102.32 | **6424.6** |
|    | 2000 | 15646.1 | 14235.5 | 6038.45 | **5489.0** |
| Average Improvement | | 20.53% | 16.40% | 23.25% | 7.74% |

Each method is independently executed three times under the 2000-sample budget. Table 12 reports the average performance, where lower values indicate better algorithm performance. Results marked with 500* are drawn from Table 9.

As shown in Table 12, with an expanded budget of 2000 samples, PartEvo maintains its lead on both benchmarks P3 and P4, consistently outperforming Funsearch, EoH, and ReEvo. On P3, its performance improves modestly from 6539.23 to 6479.49 (0.92%), while on the more challenging P4, it achieves a significant gain from 6424.6 to 5489.0 (14.4%). These results demonstrate that while PartEvo is highly efficient at discovering effective algorithms early on, it does not suffer from premature convergence. Instead, it continues to refine its search and achieve better results as more computational budget is allocated.

Meanwhile, Funsearch, EoH, and ReEvo exhibit substantial improvements with a larger budget. ReEvo benefits most, with an average improvement of 23.25%, followed by Funsearch (20.53%) and EoH (16.40%). This suggests that additional queries provide these methods with much-needed exploration capacity. Nevertheless, their performance remains far behind PartEvo. Notably, even with 2000 queries, Funsearch (15646.1) and EoH (14235.5) perform worse than PartEvo with only 500 queries (6424.6). This finding highlights PartEvo's remarkable efficiency in algorithm discovery, as it achieves up to 3–4× higher effectiveness in complex problem settings with a fraction of the computational effort.

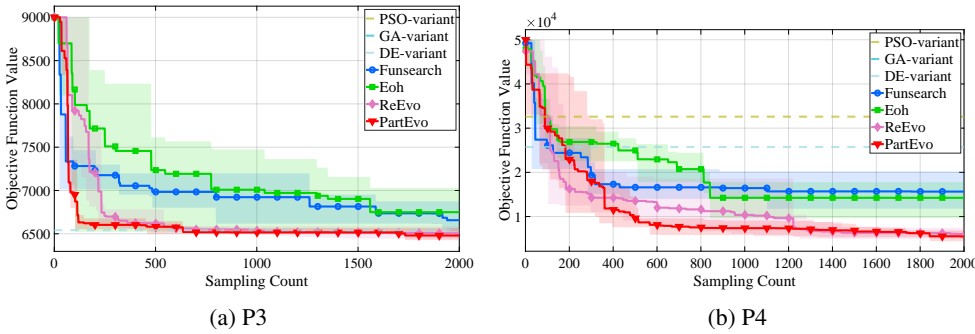

(a) P3                 (b) P4

Figure 5: Convergence behaviors of three LES methods across three runs on benchmark P3 and P4.

Figure 5 further illustrates the convergence dynamics. On P3, PartEvo surpasses the human-designed DE variant baseline with only 600 queries and continues to improve, while ReEvo catches up after about 1000 queries. Funsearch and EoH converge more slowly, indicating lower search efficiency. On P4, PartEvo again dominates the entire trajectory, discovering strong meta-heuristics as early as 500 queries and steadily improving thereafter, demonstrating resilience against local optima. ReEvo progresses more gradually, only approaching PartEvo's 600-query performance after nearly 1200 queries, while Funsearch and EoH show limited late-stage gains.

By the 500-query mark, PartEvo has already identified highly effective algorithms, leaving limited room for further improvement—a pattern that explains its relatively modest 7.7% gain when the budget increases to 2000. Importantly, this efficiency is not transient but a fundamental property of PartEvo's search process, making it a consistently robust choice across both constrained and generous budgets.

Overall, these results highlight two key findings: (i) PartEvo achieves remarkable efficiency without premature stagnation, and (ii) integrating mature EC principles into LES is crucial for sustaining effective algorithm discovery under varying computational budgets.

## B.7 Comparative evaluations with different LLMs

To investigate the influence of LLM capability on the overall performance of algorithm discovery, we evaluate PartEvo with three different models: GPT-3.5-turbo, GPT-4o-mini, and GPT-4o. Experiments are conducted on the P4 benchmark with identical initial populations, a fixed budget of 500 samples, and three independent runs for each model. For clarity of presentation, the results of the three runs are sorted in ascending order of performance. The outcomes are summarized in Table 13.

Table 13: Performance comparison of PartEvo with different LLMs on the P4 benchmark.

| LLM | Run 1 | Run 2 | Run 3 | Average | SEM |
|---|---|---|---|---|---|
| GPT-3.5-turbo | 4924.5 | 9200.2 | 13261.9 | 9128.9 | 1965.360 |
| GPT-4o-mini | 4539.1 | 4679.5 | 10055.4 | 6424.7 | 1482.610 |
| GPT-4o | 3304.6 | 6028.0 | 8187.5 | 5840.0 | 1153.466 |

Overall, PartEvo demonstrates the ability to generate high-performing algorithms across all tested LLMs. As shown in Table 13, stronger LLMs tend to yield better and more stable performance. These findings support our discussion in Appendix A.2, suggesting that as LLMs continue to advance, we can expect further improvements in PartEvo's algorithm discovery capability.

## B.8 Comparative evaluations with various clustering methods

In PartEvo, clustering serves as a tool for grouping individuals within the feature space. As we outlined in Section 3.1, PartEvo is designed to be compatible with a variety of clustering methods. We initially chose K-Means for its simplicity, transparency, and widespread use. This deliberate choice allows us to isolate and confirm that the observed performance improvements are primarily due to

the effects of niche-based evolution, rather than influenced by the complexities of more advanced clustering methods.

To further demonstrate PartEvo's flexibility and compatibility, we conducted additional experiments using three distinct clustering methods: K-Means, Gaussian Mixture Model (GMM), and Spectral Clustering (SC). Each version of PartEvo was initialized with the same feature mapping and initial population to ensure a fair comparison. Each method was run three times independently to ensure the robustness of the results. The experimental results, comparing the performance of these PartEvo variants against the baseline method EoH, are presented in Table 14.

Table 14: Performance comparison of PartEvo with different clustering methods

| Methods | P1 | P2 | P3 | P4 |
|---|---|---|---|---|
| EoH | 1.846 | 800.98 | 7268.22 | 19152.4 |
| PartEvo-K-means | 0.074 | 800.42 | 6539.24 | 6424.7 |
| PartEvo-GMM | 0.048 | 800.10 | 6580.82 | 5423.5 |
| PartEvo-SC | 0.075 | 800.01 | 6543.54 | 6714.2 |

As the results show, all PartEvo variants consistently and significantly outperform EoH across all four benchmarks, regardless of the specific clustering method used. On problems P1, P2, and P3, the performance metrics for PartEvo-K-means, PartEvo-GMM, and PartEvo-SC are remarkably similar, all achieving superior results to EoH. On the P4 problem, PartEvo-GMM shows a slight edge over the other two PartEvo variants, but this minor difference does not change the overarching conclusion that PartEvo, as a framework, is fundamentally superior to the baseline method.

These findings strongly support our claim that the PartEvo framework is compatible with various clustering algorithms. Furthermore, they provide compelling evidence that the core advantage of our approach lies in its niche-based evolutionary mechanism, rather than the specific complexities of any single clustering method. This adaptability allows practitioners to select a clustering algorithm based on the specific requirements of their application.

## C   An example of PartEvo in practice

To facilitate a better understanding of how PartEvo operates in practice, this section presents both the pseudocode and a concrete application example.

### C.1   Pseudocode of PartEvo

The pseudocode in Algorithm 1 outlines the overall procedure of PartEvo. Given a natural language description of the algorithm design problem and a set of evaluation instances, PartEvo autonomously generates customized algorithms.

---
**Algorithm 1** Pseudocode of PartEvo

---
**Require:** Task description, training instances
 1: Initialize the algorithm population
 2: Construct algorithm niches via *feature-assisted niche construction*
 3: **while** termination condition not met **do**
 4:     **for** each niche in parallel **do**
 5:         **for** each evolutionary operator **do**
 6:             Select parent algorithms
 7:             Construct few-shot prompt (parents + task description + operator-specific instructions)
 8:             Generate offspring algorithm via LLM
 9:             Evaluate offspring on training instances
10:             Manage offspring at the niche level
11:         **end for**
12:     **end for**
13: **end while**
14: **return** Best designed algorithm so far

---

In **Line 1**, the initial algorithm population is generated either using a batch initialization strategy with LLMs or by loading pre-existing algorithms. In **Line 2**, algorithm niches are constructed based on the initial population using *feature-assisted niche construction*. The process then enters an iterative evolutionary loop (**Lines 3-13**), which continues until a predefined termination condition is met, such as reaching a maximum number of generations or achieving a target fitness threshold. Within this loop, evolution occurs in parallel across each niche (**Line 4**). In each niche, designed evolutionary operators are applied sequentially or in a random order to generate new algorithms (**Lines 5-10**). Each algorithm generation step includes selecting parent algorithms, constructing prompts, generating offspring algorithms using LLMs, evaluating the offspring, and managing the individuals at the niche level. Finally, the best-performing algorithm is returned at **Line 14**.

### C.2 A concrete example on benchmark P1

As an example, consider Benchmark P1, where the goal is to discover meta-heuristic algorithms for unimodal optimization problems. PartEvo initiates the process by defining the task description and preparing a set of training instances. The task description specifies the algorithm design problem, guiding PartEvo in generating relevant algorithms, as depicted in Figure 14. The training instances serve as practice problems, providing a consistent basis for evaluating and ranking candidate algorithms during the evolutionary process. These instances are representative of the problem domain and help refine the algorithms' effectiveness on the task. In the case of P1, the goal is to solve unimodal optimization problems, for which standard unimodal optimization functions are chosen as the target functions.

Subsequently, PartEvo proceeds through iterative evolutionary loops. In each loop iteration, parent algorithms are selected, prompts are constructed, new offspring algorithms are generated by the LLM, and these offspring algorithms are evaluated and managed. The prompt construction follows the template described in Appendix D. Through repeated iterations, the algorithm population gradually improves, with the best-performing algorithms emerging as specialized solutions to unimodal optimization problems.

This workflow generalizes beyond unimodal optimization. For any benchmark or real-world application, one only needs to specify the problem and provide representative training instances; **PartEvo** will then evolve tailored algorithms accordingly.

## D Prompts

This section describes the prompts employed in PartEvo. In all displayed prompts, the black text remains fixed, while the red placeholders correspond to task-specific elements, and the blue placeholders change throughout the evolutionary process.

Figures 6 and 7 show the prompts used by the summarizer and reviewer LLMs. The generator LLMs, which utilize different prompts for each of the four evolutionary operators (e.g., RE, SE, CN, LGE), are detailed in Figures 10 to 13. These prompts are applicable across all four optimization problem types.

Task description prompts for the four problem types are shown in Figure 14. Specifically, all problems are presented as black-box optimization tasks to the LLMs. This approach ensures that PartEvo is effective across a wide range of optimization problems.

## E Feature projection

The general LES framework employs an abstract search space partitioning mechanism that projects sampled individuals into a feature space. In this feature space, clustering is performed based on the distances between the mapped representations, which indirectly partitions the abstract search space into multiple subregions. PartEvo leverages two primary methods for feature mapping:

### E.1 Code similarity vector

Similar individuals should exhibit comparable relative distances to all other individuals. Thus, we construct a Code Similarity Vector (CSV) for each individual to represent its similarity to all

You are the person responsible for recording the progress of the experts' research. Some experts are working on the following task:
**"[Task Description Placeholder]"**
Based on previous design methods, we have the following summary:
**"[Previous Summary Placeholder]"**
Experts have now explored an additional **[Number Placeholder]** algorithms to tackle this problem, ranging from No. 1 to No. **[Number Placeholder]**, with performance gradually declining (the smaller the objective function value, the better the performance).
Algorithm No. 1 has an objective function value of **[Value Placeholder]**, and its idea is:
**"[Thought Placeholder]"**
Algorithm No. 2 has an objective function value of **[Value Placeholder]**, and its idea is:
**"[Thought Placeholder]"**
...
Please review the previous experiences and the current algorithms, analyzing which techniques within the algorithms are effective in solving this problem and which are not. Finally, summarize both the effective and ineffective techniques, and update the previous summary accordingly. Please ensure that the summary you provide is enclosed in braces.

Figure 6: Prompt of summarizer.

Figure 7: Prompt of reviewer.

Figure 8: Prompt of generator for initiation.

others. As a result, the distances between CSVs of similar individuals should be small. Since code is generated by LLMs, there may be many implementations of functionally equivalent code. In constructing the CSV, we focus on both the syntactic and dataflow similarities between codes. The process of constructing the CSV is as follows:

1. **Pairwise similarity calculation**: For a population of size $N$, we compute the pairwise similarity between all individuals and store the results in a similarity matrix $S$. Specifically, we use the

You are an algorithm design expert. An intelligent agent is currently executing the following design task:
**"[Task Description Placeholder]"**
The agent has designed an algorithm with the following ideas and code:
**"[Thought Placeholder]"**
**"[Code Placeholder]"**
An expert has provided some suggestions for this algorithm. You can decide whether to incorporate the expert's feedback, and then create a new algorithm that differs from the given one but motivated by it. The suggestion is: **"[Reflection Placeholder]"**
First, describe your concept for the new algorithm and its main steps in as few words as possible while ensuring clarity. The description must be enclosed in braces. Next, implement it in Python as a runnable function named **"[Function Signature Placeholder]"**. This function should accept **"[Value Placeholder]"** input(s): **"[Input Placeholder]"**. The function should return **"[Value Placeholder]"** output(s): **"[Output Placeholder]"**. You can understand the inputs and outputs based on the current algorithm's code. Do not include any comments in the code.

Figure 9: Prompt of generator in RE.

You are an algorithm design expert. An intelligent agent is currently executing the following design task:
**"[Task Description Placeholder]"**
The agent has designed an algorithm with the following ideas and code:
**"[Thought Placeholder]"**
**"[Code Placeholder]"**
Please help me create a new algorithm that is different from the given one but motivated by it.
First, describe your concept for the new algorithm and its main steps in as few words as possible while ensuring clarity. The description must be enclosed in braces. Next, implement it in Python as a runnable function named **"[Function Signature Placeholder]"**. This function should accept **"[Value Placeholder]"** input(s): **"[Input Placeholder]"**. The function should return **"[Value Placeholder]"** output(s): **"[Output Placeholder]"**. You can understand the inputs and outputs based on the current algorithm's code. Do not include any comments in the code.

Figure 10: Prompt of generator in Mutation.

You are an algorithm design expert, currently collaborating with other experts on the following task:
**"[Task Description Placeholder]"**
Experts have designed **"[Value Placeholder]"** algorithms with their corresponding codes.
The No. 1 algorithm and the corresponding code are:
**"[Thought Placeholder]"**
**"[Code Placeholder]"**
The No. 2 algorithm and the corresponding code are:
**"[Thought Placeholder]"**
**"[Code Placeholder]"**
...
Please take Algorithm No. 1 as the main framework and try to incorporate the characteristics of the other algorithms into it to create a better algorithm. First, describe your concept for the new algorithm and its main steps in as few words as possible while ensuring clarity. The description must be enclosed in braces. Next, implement it in Python as a runnable function named **"[Function Signature Placeholder]"**. This function should accept **"[Value Placeholder]"** input(s): **"[Input Placeholder]"**. The function should return **"[Value Placeholder]"** output(s): **"[Output Placeholder]"**. You can understand the inputs and outputs based on the current algorithm's code. Do not include any comments in the code.

Figure 11: Prompt of generator in CN.

CodeBLEU metric [50] to evaluate the similarity between the code of two individuals. CodeBLEU

You are an algorithm design expert, currently collaborating with other experts on the following task:
**"[Task Description Placeholder]"**
Currently, **"[Value Placeholder]"** algorithms have been explored for this problem, with their effectiveness decreasing from No. 1 to No. **"[Value Placeholder]"**. The concepts for these methods are as follows. No. 1 algorithm is: **"[Thought Placeholder]"**
No. 2 algorithm is: **"[Thought Placeholder]"**

...

Please analyze the summary and then modify the following algorithm to create a more promising solution for this problem. The thoughts and code for the algorithm to be modified are as follows:
**"[Thought Placeholder]"**
**"[Code Placeholder]"**
First, describe your concept for the new algorithm and its main steps in as few words as possible while ensuring clarity. The description must be enclosed in braces. Next, implement it in Python as a runnable function named **"[Function Signature Placeholder]"**. This function should accept **"[Value Placeholder]"** input(s): **"[Input Placeholder]"**. The function should return **"[Value Placeholder]"** output(s): **"[Output Placeholder]"**. You can understand the inputs and outputs based on the current algorithm's code. Do not include any comments in the code.

Figure 12: Prompt of generator in SE.

You are an algorithm design expert, currently collaborating with other experts on the following task:
**"[Task Description Placeholder]"**
Experts are divided into several groups, with each group responsible for the development of a specific algorithm cluster. Each cluster incorporates different techniques while maintaining its own framework to explore diverse algorithms.
On your algorithm cluster, after several iterations, the current algorithm (idea and the corresponding code) is:
**"[Thought Placeholder]"**
**"[Code Placeholder]"**
During the iterations in your cluster, a better-performing algorithm appeared, and its idea and code are as follows:
**"[Thought Placeholder]"**
**"[Code Placeholder]"**
In addition, among all the algorithms tested (including those from other clusters), the best-performing algorithm's idea and code are as follows:
**"[Thought Placeholder]"**
**"[Code Placeholder]"**
Using the above information and adhering to the core framework of the current algorithm, please suggest potential improvements to enhance its performance in solving this problem. First, briefly describe your concept for the new algorithm and its main steps. The description must be enclosed in braces. Next, implement it in Python as a function named **"[Function Signature Placeholder]"**. This function should accept **"[Value Placeholder]"** input(s): **"[Input Placeholder]"**. The function should return **"[Value Placeholder]"** output(s): **"[Output Placeholder]"**. You can understand the inputs and outputs based on the current algorithm's code. Do not include any comments in the code.

Figure 13: Prompt of generator in LGE.

proposes weighted n-gram match and syntactic abstract syntax tree match to measure grammatical correctness, and introduces semantic data-flow match to calculate logic correctness. In this work, the similarity score between two individuals $i$ and $j$ is computed as:

$$S_{i,j} = 0.5 \cdot s_{i,j} + 0.5 \cdot d_{i,j} \tag{1}$$

where $s_{i,j}$ and $d_{i,j}$ denote the syntax match score and dataflow match score, respectively. This formulation ensures a balanced evaluation of both syntactic structure and semantic behavior.

You are tasked with solving a black-box **optimization problem/Mixed-Integer Nonlinear Programming (MINLP) problem/integer nonlinear programming (INLP)** using meta-heuristic optimization algorithm. The goal is to minimize the objective value. Please design a metaheuristic algorithm that can effectively solve this optimization problem and return the optimal solution.

Figure 14: Prompt of specific problem.

2. **Symmetrization**: CodeBLEU scores are not inherently symmetric, i.e., $S_{i,j} \neq S_{j,i}$, which may lead to inconsistencies in the similarity matrix. To address this, we enforce symmetry by averaging the scores:

$$S = \frac{S + S^\top}{2}. \tag{2}$$

This step ensures that the similarity relationship between any two individuals is consistent and interpretable.

3. **Feature representation**: For each individual, the corresponding row of the similarity matrix is used as its CSV, capturing its relative relationship with all other individuals in the population.

### E.2 Thought embedding

In PartEvo, individuals are encoded into two components: code and thought. For the thought component, expressed in natural language, we leverage techniques from natural language processing to perform feature projection. Specifically, we utilize pre-trained language models, such as BERT [63], to extract high-dimensional semantic representations of the algorithmic descriptions provided by individuals. The similarity between individuals is then determined based on the distances between their embedding vectors. The process is as follows:

1. **Text tokenization**: The algorithmic description of each individual is tokenized using the BERT tokenizer, ensuring truncation and padding to a maximum sequence length of 512 tokens.
2. **Embedding extraction**: The tokenized input is fed into a pre-trained BERT model, and the hidden state of the token from the final layer is extracted as the embedding. This embedding is a fixed-length vector capturing the semantic meaning of the input text.
3. **Feature representation**: The extracted embeddings are directly used as feature vectors for the individuals.

### E.3 Random features for comparison

To evaluate the effectiveness of the adopted feature projections, we also introduce random feature vectors as a baseline. For each individual, a random vector is generated by sampling from a Gaussian distribution with a dimensionality consistent with the other feature representations. As shown in Table 5, the comparison with random features demonstrates that the proposed feature projections significantly enhance the performance of PartEvo. This improvement is one of the key reasons why PartEvo achieves higher search efficiency compared to Funsearch, which employs an island model.

## F Human-designed meta-heuristic algorithms

In this work, we use human-designed meta-heuristic algorithms as baselines. Specifically, we adopt enhanced variants of Genetic Algorithm (GA), Differential Evolution (DE), and Particle Swarm Optimization (PSO). Detailed descriptions of these variants are provided below.

### F.1 GA-variant

GA is a classical meta-heuristic optimization algorithm inspired by the process of natural selection. It iteratively evolves a population of candidate solutions through genetic operations such as selection, crossover, and mutation. The GA-variant used in this work incorporates several enhancements to improve its performance and convergence behavior. These enhancements include:

- **Selection:** A hybrid selection mechanism combining Roulette Wheel Selection (RWS) and Tournament Selection (TS) is employed. TS is used as the primary selection method, with a tournament size of 3, while RWS ensures diversity preservation.

- **Crossover:** Simulated Binary Crossover (SBX) is adopted as the primary crossover operator. SBX generates offspring by simulating the distribution of genes in natural reproduction, with a crossover rate of 0.9 and a distribution index $\eta = 2$. Single-point crossover is also supported as a fallback.

- **Mutation:** Gaussian Mutation and Uniform Mutation are used to introduce diversity into the population. Gaussian Mutation adds normally distributed noise to selected genes, with a mutation rate of 0.1.

- **Fitness normalization:** To mitigate the effects of extreme fitness values, a normalization step is applied to the fitness scores before selection.

- **Elitism:** The best solution in the current generation is preserved and directly passed to the next generation, ensuring that the global best solution is not lost.

- **Early stopping:** An early stopping mechanism is implemented to terminate the algorithm if no significant improvement in fitness is observed for 500 consecutive generations. The convergence tolerance is set to $10^{-6}$.

The GA variant is designed to balance exploration and exploitation effectively. The use of hybrid selection and advanced crossover/mutation operators ensures that the algorithm can explore the search space thoroughly, while elitism and early stopping mechanisms help accelerate convergence and avoid stagnation.

### F.2   DE-variant

DE is a population-based optimization algorithm that relies on mutation, crossover, and selection operators to iteratively improve candidate solutions. The DE variant used in this work incorporates several enhancements to improve its convergence speed, robustness, and ability to escape local optima. These enhancements include:

- **Adaptive mutation factor and crossover probability:** The mutation factor $F$ and crossover probability $CR$ are dynamically adjusted during the optimization process. Specifically, $F$ and $CR$ are sampled from uniform distributions in the ranges $[0.5, 0.8]$ and $[0.5, 0.9]$, respectively, to balance exploration and exploitation.

- **Global best participation:** The global best solution is incorporated into the mutation process to guide the search towards promising regions of the solution space. This modification improves convergence by leveraging the knowledge of the current best solution.

- **Diversity maintenance:** To avoid premature convergence, random perturbations are introduced when population diversity is low. This helps the algorithm escape local optima and explore new regions of the search space.

- **Boundary handling:** A reflection boundary handling strategy is used to prevent solutions from being stuck at the boundaries of the search space. This ensures that mutated solutions remain within valid bounds while preserving diversity.

- **Early stopping:** An early stopping mechanism is implemented to terminate the algorithm when no significant improvement in fitness is observed for 500 consecutive generations. The convergence tolerance is set to $10^{-6}$.

- **Parallel evaluation:** For computational efficiency, the evaluation of the objective function is parallelized when possible, particularly for computationally expensive objective functions.

The DE variant is designed to enhance the standard DE algorithm's performance in solving complex optimization problems. By incorporating adaptive parameters, diversity maintenance, and boundary handling, it achieves a good balance between exploration and exploitation. The early stopping mechanism further improves computational efficiency by avoiding unnecessary evaluations when the algorithm has converged.

### F.3 PSO-variant

Particle Swarm Optimization (PSO) is a population-based optimization algorithm inspired by the social behavior of bird flocks and fish schools. In PSO, particles explore the search space by updating their positions and velocities based on their own best-known positions and the global best-known position. The PSO variant used in this work integrates several advanced mechanisms to enhance its convergence performance and robustness. These enhancements include:

- **Dynamic inertia weight:** The inertia weight $w$ is linearly decreased from $w_{\max} = 0.95$ to $w_{\min} = 0.4$ over the course of the optimization to balance global exploration and local exploitation.

- **Simulated Annealing (SA):** A simulated annealing mechanism is incorporated to probabilistically accept worse solutions during the optimization process. This helps the algorithm escape local optima, with the acceptance probability controlled by a temperature parameter that decreases exponentially.

- **Genetic Algorithm (GA) operators:** Genetic algorithm-inspired operators, including crossover and mutation, are integrated to further enhance diversity. Specifically:

  - *Crossover:* Offspring particles are generated by combining the local best positions and the global best position using a weighted random combination.
  - *Mutation:* A small mutation probability ($p_m = 0.02$) is used to introduce random perturbations to particles.

- **Boundary handling:** A reflection boundary handling strategy is applied to ensure particles remain within the feasible search space. This prevents particles from being stuck at the boundaries while maintaining diversity.

- **Velocity clamping:** The velocity of each particle is clamped to prevent it from exceeding the maximum allowable velocity, which is proportional to the search space bounds.

- **Hybrid fitness evaluation:** The fitness of each particle is evaluated using a combination of PSO updates, GA operators, and simulated annealing, ensuring a robust exploration of the search space.

The PSO variant combines elements from PSO, GA, and SA, making it a hybrid algorithm that leverages the strengths of each method. The dynamic inertia weight helps balance exploration and exploitation, while the simulated annealing and genetic operators enhance the algorithm's ability to escape local optima. These modifications make the PSO variant well-suited for solving complex optimization problems.

### F.4 Comparison and analysis

To directly compare the performance of the standard and enhanced versions of GA, DE, and PSO, we evaluated all methods on four benchmark problems (P1-P4), as summarized in Table 15. In this table, GA, DE, and PSO refer to the standard versions, while GA-variant, DE-variant, and PSO-variant represent the enhanced versions discussed in the main text. Each algorithm was limited to a maximum of 30,000 solution evaluations. The best results are highlighted in bold, while the second-best results are underlined.

Table 15: Best results from multiple runs on training and testing instances for each optimization benchmark (Lower values indicate better performance)

| Problem | Instances | GA | GA-variant | DE | DE-variant | PSO | PSO-variant | PartEvo |
|---------|-----------|-----|-----------|-----|-----------|-----|------------|---------|
| P1 | Training | $5.944 \times 10^6$ | 523.182 | 15.849 | 0.004 | 118.973 | 16.241 | **0.000** |
|    | Testing | $6.358 \times 10^{22}$ | 13.145 | 0.680 | **0.000** | 50.000 | **0.000** | **0.000** |
| P2 | Training | 2485.72 | 2087.63 | 1139.32 | 868.40 | 1467.15 | 939.02 | **800.00** |
|    | Testing | $1.05 \times 10^5$ | 542.48 | 516.96 | 365.60 | $2.11 \times 10^4$ | 411.80 | **344.62** |
| P3 | Training | 15661.66 | 52850.88 | 21379.04 | 6541.55 | 156817.83 | 10036.17 | **6471.31** |
|    | Testing | 78080.20 | 85629.55 | 77851.57 | 61298.43 | $2.89 \times 10^5$ | 60464.14 | **56876.40** |
| P4 | Training | $2.60 \times 10^6$ | $1.48 \times 10^7$ | 91784.6 | 25736.4 | 82953.9 | 32594.0 | **2792.1** |
|    | Testing | $5.13 \times 10^7$ | $8.48 \times 10^7$ | $3.78 \times 10^5$ | $1.72 \times 10^5$ | $6.31 \times 10^6$ | $2.28 \times 10^5$ | **14396.1** |

From the table, several key observations can be made. **First**, the performance of different algorithms varies significantly for each problem, highlighting the distinct behaviors of each algorithm within the search space, which allows them to adapt to different landscapes.

**Second**, within each family of algorithms, the enhanced variants typically outperform the standard versions, except for the GA variants on problems P3 and P4. This suggests that targeted algorithmic improvements enable more effective navigation of the solution landscape. However, improper upgrades can lead to performance degradation, underscoring the importance of thoughtful algorithm design.

**Most notably**, the algorithms generated by PartEvo achieve superior performance across all benchmark instances. This is because PartEvo continuously explores techniques that are well-suited to the problem during the evolutionary process, allowing the designed algorithms to efficiently search the landscape. This demonstrates that PartEvo plays a crucial role in automating and enhancing the algorithm design process, highlighting its potential to assist human experts in developing high-performance optimization algorithms.

## G   Benchmark

This section introduces the four optimization benchmarks and their specific instances used to evaluate PartEvo. These include synthetic problems, such as unimodal and multimodal optimization problems, as well as real-world scenarios, including task offloading in mobile edge computing systems and machine-level scheduling for heterogeneous plants.

### G.1   Unimodal optimization problem

Unimodal optimization problems feature a single global optimum with no local optima, making it an ideal benchmark for testing basic search capabilities. Algorithms yielded by the LES framework are evaluated on their ability to locate the global optimum in this simple scenario efficiently. This benchmark serves as a baseline to verify PartEvo's ability to generate algorithms with effective core search mechanisms.

For this study, we utilize various unimodal objective functions. For some of these functions, we introduce a shift term $\mathbf{s}$. This shift increases the difficulty of optimization, preventing the LES framework from generating algorithms with fixed outputs that could bypass the optimization process. In the experiments, the parameter configurations for the training and testing instances are shown in Table 16. The mathematical formulations and key characteristics of these ten functions are outlined as follows:

Table 16: Unimodal instance setups for the experiments

|  | Formula | Variable Range | Optimal Function Value |
|---|---|---|---|
| Training Instance | $f_1(\mathbf{x})$ | [-20, 20] | 0 |
| | $f_2(\mathbf{x})$ | [-20, 20] | 0 |
| | $f_2(\mathbf{x}, \mathbf{s} = 3)$ | [-20, 20] | 0 |
| Testing Instance | $f_2(\mathbf{x}, \mathbf{s} = 2.5)$ | [-10, 10] | 0 |
| | $f_2(\mathbf{x}, \mathbf{s} = 5)$ | [-10, 10] | 0 |
| | $f_3(\mathbf{x}, \mathbf{s} = 0)$ | [-10, 10] | 0 |
| | $f_3(\mathbf{x}, \mathbf{s} = 2.5)$ | [-5, 5] | 0 |
| | $f_4(\mathbf{x}, \mathbf{s} = 1)$ | [-30, 30] | $\approx 0$ |
| | $f_5(\mathbf{x})$ | [-100, 100] | $\approx 0$ |
| | $f_6(\mathbf{x})$ | [-100, 100] | $\approx 0$ |

**Sphere function.** The Sphere function is a well-known unimodal optimization benchmark, characterized by a global minimum located at the origin. It is mathematically defined as:

$$f_1(\mathbf{x}) = \sum_{i=1}^{n} x_i^2,\tag{3}$$

where $\mathbf{x} = [x_1, x_2, \ldots, x_n] \in \mathbb{R}^n$ represents the decision variable in an $n$-dimensional space. The function exhibits a smooth, symmetric parabolic shape, with the global minimum located at $\mathbf{x} = 0$. This simplicity makes it a fundamental benchmark for evaluating optimization algorithms.

**Shifted sphere function.** The Shifted Sphere function is a modified version of the Sphere function, where the global minimum is shifted to a predefined point in the search space. This shift increases the difficulty of the optimization process, preventing trivial solutions. It is mathematically expressed as:

$$f_2(\mathbf{x}, \mathbf{s}) = \sum_{i=1}^{n} (x_i - s_i)^2, \tag{4}$$

where $\mathbf{s} = [s_1, s_2, \ldots, s_n] \in \mathbb{R}^n$ represents the shift vector, and $s_i$ denotes the shift applied to the $i$-th dimension.

**Unimodal sinusoidal function.** The Unimodal Sinusoidal function is a single-peaked benchmark, characterized by the use of squared sine terms. It is mathematically defined as:

$$f_3(\mathbf{x}, \mathbf{s}) = \sum_{i=1}^{n} \sin^2(x_i - s_i) \tag{5}$$

This function is particularly useful for testing optimization algorithms on periodic landscapes, as its sinusoidal nature introduces oscillatory behavior while maintaining a single global minimum, making it a valuable benchmark for evaluating algorithm robustness.

**Unimodal Gaussian function.** The Unimodal Gaussian function is a smooth and symmetric benchmark function, characterized by a single peak. It is mathematically expressed as:

$$f_4(\mathbf{x}, \mathbf{s}) = \sum_{i=1}^{n} \exp\left(-(x_i - s_i)^2\right), \tag{6}$$

here, $\mathbf{s} = [s_1, s_2, \ldots, s_n] \in \mathbb{R}^n$ is the shift vector that determines the location of the global minimum. This function is often used in optimization benchmarks due to its smooth, unimodal nature and its ability to test the precision of optimization algorithms.

**Unimodal logistic function.** The Unimodal Logistic function is a smooth, monotonic benchmark function, commonly used to evaluate optimization algorithms in continuous domains. It is mathematically defined as:

$$f_5(\mathbf{x}) = \sum_{i=1}^{n} \frac{1}{1 + \exp(-x_i)}, \tag{7}$$

This function is particularly useful for testing algorithms in scenarios involving smooth, non-linear transformations.

**Unimodal exponential function.** The Unimodal Exponential function is a monotonic benchmark function characterized by exponential decay. It is mathematically expressed as:

$$f_6(\mathbf{x}) = \sum_{i=1}^{n} \exp(-x_i), \tag{8}$$

The global minimum of this function is asymptotically approached as each $x_i$ tends towards positive infinity, making it a useful benchmark for testing optimization algorithms in decaying landscapes.

### G.2  Multimodal optimization problem

Multimodal optimization problems introduce multiple local optima in the search space, making them significantly more challenging. These problems evaluate the designed algorithms' ability to balance exploration and exploitation. Practical algorithms must avoid premature convergence to suboptimal solutions while maintaining a robust search strategy. This benchmark tests PartEvo's ability to address complex optimization landscapes.

We utilize ten multimodal objective functions in our study. Similar to the unimodal optimization benchmarks, we introduce a shift term $\mathbf{s}$ in certain functions to increase the difficulty of optimization. The parameter configurations for the training and testing instances are summarized in Table 17.

Specifically, the decision variable dimensionality for the training instances is set to 40, while that for the testing instances is set to 50. This setup allows us to evaluate the generalization capability of the metaheuristic algorithms generated by LESs. The mathematical formulations and key characteristics of these functions are outlined below.

Table 17: Multimodal instance setups for the experiments

|  | Formula | Variable Range | Optimal Function Value |
|---|---|---|---|
| Training Instance | $f_7(\mathbf{x}, \mathbf{s} = 2)$ | [-5.12, 5.12] | 100 |
|  | $f_9(\mathbf{x}, \mathbf{s} = 50)$ | [-600, 600] | 200 |
|  | $f_{10}(\mathbf{x}, \mathbf{s} = 5)$ | [-10, 10] | 500 |
| Testing Instance | $f_8(\mathbf{x}, \mathbf{s} = 10)$ | [-10, 10] | 50 |
|  | $f_{11}(\mathbf{x})$ | [-100, 100] | 300 |
|  | $f_{12}(\mathbf{x})$ | [-0.5, 0.5] | 10 |
|  | $f_{13}(\mathbf{x})$ | [-10, 10] | 0 |
|  | $f_{14}(\mathbf{x})$ | [0, $\pi$] | -50 |

**Modified Rastigin function.** The Modified Rastrigin function is a multimodal benchmark with numerous local minima, making it a popular choice for evaluating optimization algorithms. It is mathematically defined as:

$$f_7(\mathbf{x}, \mathbf{s}) = A \cdot n + \sum_{i=1}^{n} \left[ (x_i - s_i)^2 - A \cdot \cos(2\pi(s_i)) \right] + 100, \tag{9}$$

where $A = 10$, and the search range is $x_i \in [-5.12, 5.12]$. The global minimum is located at $\mathbf{x}^* = \mathbf{s}$, with an optimal value of $f(\mathbf{x}^*) = 100$.

**Modified Ackley function.** The Modified Ackley function is a widely used multimodal benchmark with a complex landscape. It is mathematically expressed as:

$$f_8(\mathbf{x}, \mathbf{s}) = -20 \exp\left(-0.2 \sqrt{\frac{1}{n} \sum_{i=1}^{n} (x_i - s_i)^2}\right) - \exp\left(\frac{1}{n} \sum_{i=1}^{n} \cos(2\pi(x_i - s_i))\right) + 20 + e + 50, \tag{10}$$

where $e$ is Euler's constant, and the search range is $x_i \in [-32.768, 32.768]$. The global minimum is shifted to $\mathbf{x}^* = \mathbf{s}$, with an optimal value of $f(\mathbf{x}^*) = 50$.

**Modified Griewank function.** The Modified Griewank function features a smooth landscape with numerous local minima. It is mathematically defined as:

$$f_9(\mathbf{x}, \mathbf{s}) = 1 + \frac{1}{4000} \sum_{i=1}^{n} (x_i - s_i)^2 - \prod_{i=1}^{n} \cos\left(\frac{x_i - s_i}{\sqrt{i}}\right) + 200. \tag{11}$$

The search range is $x_i \in [-600, 600]$. The global minimum is located at $\mathbf{x}^* = \mathbf{s}$, with an optimal value of $f(\mathbf{x}^*) = 200$.

**Modified Levy function.** The Modified Levy function is known for its rugged landscape and numerous local minima. It is mathematically expressed as:

$$f_{10}(\mathbf{x}, \mathbf{s}) = \sin^2(\pi w_1) + \sum_{i=1}^{n-1} \left[ (w_i - 1)^2 \left(1 + 10 \sin^2(\pi w_i + 1)\right) \right] + (w_n - 1)^2 \left(1 + \sin^2(2\pi w_n)\right) + 500, \tag{12}$$

where $w_i = 1 + \frac{x_i - s_i}{4}$, and the search range is $x_i \in [-10, 10]$. The global minimum is located at $\mathbf{x}^* = s_i$, with an optimal value of $f(\mathbf{x}^*) = 500$.

**Modified Schaffer function N.2.** The Modified Schaffer Function N.2 is a two-dimensional multimodal benchmark. It is mathematically expressed as:

$$f_{11}(\mathbf{x}) = 0.5 + \frac{\sin^2\left((x_1 - 50)^2 - (x_2 + 50)^2\right) - 0.5}{\left[1 + 0.001\left((x_1 - 50)^2 + (x_2 + 50)^2\right)\right]^2} + 300. \tag{13}$$

The search range is $x_i \in [-100, 100]$, and the global minimum is located at $\mathbf{x}^* = [50, -50]$, with an optimal value of $f(\mathbf{x}^*) = 300$.

**Modified Weierstrass function.** The Modified Weierstrass function is characterized by its fractal-like landscape. It is mathematically expressed as:

$$f_{12}(\mathbf{x}) = \sum_{i=1}^{n} \sum_{k=0}^{k_{\max}} a^k \cos\left(2\pi b^k (x_i + 0.5)\right) - n \sum_{k=0}^{k_{\max}} a^k \cos\left(2\pi b^k \cdot 0.5\right) + 10, \tag{14}$$

where $a = 0.5$, $b = 3$, $k_{\max} = 20$, and the search range is $x_i \in [-0.5, 0.5]$. The global minimum is located at $\mathbf{x}^* = [0, 0, \ldots, 0]$, with an optimal value of $f(\mathbf{x}^*) = 10$.

**Modified Alpine function.** The Modified Alpine function is a multimodal benchmark, defined as:

$$f_{13}(\mathbf{x}) = \sum_{i=1}^{n} |x_i \sin(x_i) + 0.1 x_i|. \tag{15}$$

The search range is $x_i \in [-10, 10]$, and the global minimum is located at $\mathbf{x}^* = [0, 0, \ldots, 0]$, with an optimal value of $f(\mathbf{x}^*) = 0$.

**Modified Michalewicz function.** The Modified Michalewicz function is highly multimodal, defined as:

$$f_{14}(\mathbf{x}) = -\sum_{i=1}^{n} \sin(x_i) \left[ \sin\left(\frac{i x_i^2}{\pi}\right) \right]^{2m}, \tag{16}$$

where $m = 10$, and the search range is $x_i \in [0, \pi]$. The global minimum depends on the dimensionality, with an approximate optimal value of $f(\mathbf{x}^*) \approx -n$.

### G.3 Task offloading in mobile edge computing systems

Task offloading is a well-established optimization problem in the domains of cloud and edge computing [44]. A representative system architecture is illustrated in Fig. 15. In such systems, numerous smart mobile devices are required to execute computation- and data-intensive applications. However, due to physical limitations, such as restricted device size, these devices often lack sufficient computational resources and energy to process tasks independently. Task offloading addresses this limitation by partitioning computational tasks and offloading them to high-performance servers, either at the edge or in the cloud, for efficient processing.

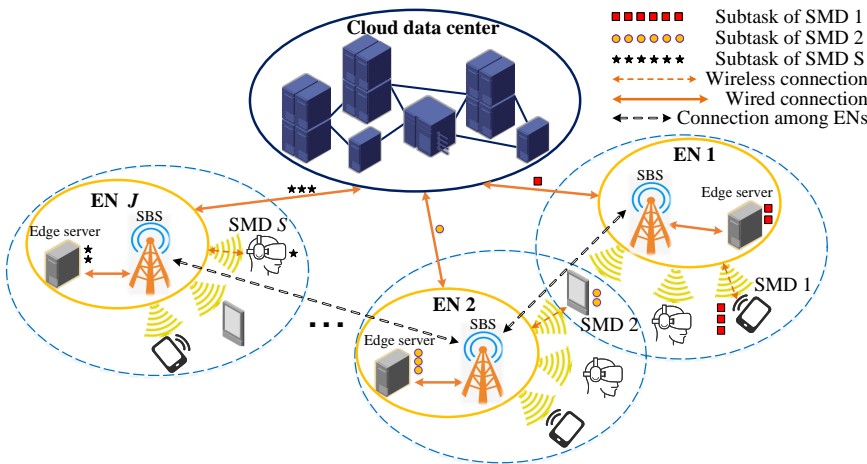

Figure 15: System architecture of a cloud-edge collaborative computing framework

The task offloading problem is inherently constrained by various practical factors, including the characteristics of application tasks, the availability of communication channels, and the computational capacity of edge servers [64]. Consequently, task offloading is frequently formulated as a constrained mixed-integer non-linear programming (MINLP) problem. The objective typically involves minimizing the total system cost, maximizing data throughput, or reducing energy consumption, all while

adhering to the imposed constraints. To tackle these complex problems, meta-heuristic algorithms are widely employed in the literature.

In this study, we treat task offloading as a real-world optimization problem to evaluate the algorithmic capabilities of PartEvo. The performance of the algorithms designed within LESs on this benchmark reflects their ability to address real-world optimization problems with intricate constraints and real-time requirements.

To this end, we implement a task offloading system model inspired by prior studies [44, 65]. The optimization objective is to minimize the total system cost. To simulate diverse real-world scenarios, we design multiple instances by varying the number of users, user request types, and user distributions. The configurations of the instances used for training and testing are summarized in Table 18.

Table 18: Configurations of task offloading instances for experiments

|  | Instance Index | Number of Users | Average Task Size (GB) |
|---|---|---|---|
| Training Instances | 1 | 15 | 150 |
| | 2 | 30 | 150 |
| Testing Instances | 3 | 5 | 50 |
| | 4 | 15 | 300 |
| | 5 | 45 | 10 |
| | 6 | 20 | 0 |
| | 7 | 30 | 200 |
| | 8 | 1 | 100 |

## G.4 Machine-Level scheduling for heterogeneous plants

Scheduling and production planning are among the most critical challenges in industrial manufacturing, as they directly impact the efficiency and cost-effectiveness of operations. In a complete production supply chain, from the moment a customer places an order, every stage—ranging from raw material procurement to the production of semi-finished components and the final assembly of diverse products—requires meticulous scheduling and coordination. Under the paradigm of industrial Internet, the entire supply chain can be modeled as a collaborative manufacturing and scheduling (CMS) system, composed of multiple heterogeneous factories. The architecture of such a system is illustrated in Figure 16.

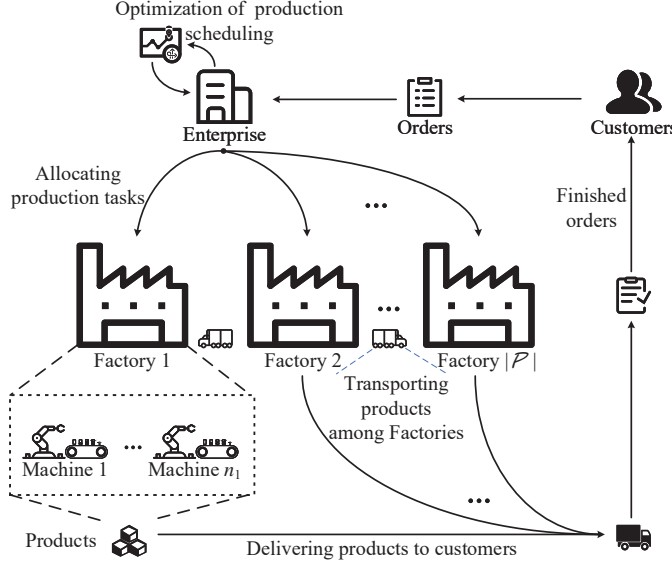

Figure 16: Architecture of a collaborative manufacturing and scheduling system.

The CMS framework involves four primary entities: enterprises, customers, a transportation system, and multiple factories. Within the system, there are $|\mathcal{P}|$ factories, and each factory $p$ ($1 \leq p \leq |\mathcal{P}|$) contains $n_p$ machines. Raw materials, semi-finished products, and finished goods are transported among factories and delivered to customers via the transportation system. The workflow proceeds as follows: a customer places an order with the enterprise, which then formulates a manufacturing and scheduling plan based on the order's requirements, including product types, quantities, and delivery deadlines. Subsequently, the enterprise assigns production tasks to machines across different factories and coordinates the transportation of products. Finally, the transportation system delivers the completed products directly to customers, fulfilling the order. To minimize the overall cost of the CMS system, it is essential to optimize the collaboration among factories, encompassing manufacturing, transportation, and sales processes.

In this context, we use the term "product" to broadly refer to all production components, including raw materials, semi-finished products, and finished goods. The production scheduling process consists of several interdependent plans: (1) production plans for different factories, (2) substitution plans for interchangeable products, (3) transportation plans for products between factories, and (4) delivery plans specifying the number of products shipped from each factory to customers. Notably, the factories within the CMS system exhibit heterogeneous characteristics, which introduce additional complexities:

1. Heterogeneous manufacturing capacities: Different factories have varying production capabilities for different products, resulting in diverse time and material costs for manufacturing the same product.

2. Variable transportation costs: Due to differing distances between factories and between factories and customers, transportation costs vary significantly across the system.

3. Complex product relationships: Certain products can substitute for others (e.g., a large resistor can be replaced by two smaller resistors in series), while others may be consumed as intermediate inputs in the production of other products. Properly managing these dependencies is critical for cost reduction.

The objective of the scheduling process is to allocate production tasks to machines and coordinate the inter-factory transportation of intermediate products. The goal is to minimize the combined production and transportation costs while ensuring that all customer orders are completed within their respective deadlines. This problem represents a large-scale combinatorial optimization challenge, often formulated as a constrained nonlinear integer programming problem. Numerous studies have explored the use of meta-heuristic algorithms to address such production scheduling problems.

In this study, we evaluate the performance of our proposed framework, PartEvo, on this industrial scheduling benchmark. PartEvo is designed to generate meta-heuristics capable of addressing large-scale, high-dimensional optimization problems with complex interdependencies. By benchmarking PartEvo on this real-world problem, we aim to assess its ability to develop efficient and effective scheduling strategies.

Table 19: Configurations of machine-level scheduling instances for experiments

|  | Instance Index | Product Types | Factory Number | Planning Horizon |
|---|---|---|---|---|
| Training Instances | 1 | 9 | 3 | 7 days |
| Testing Instances | 2 | 9 | 3 | 7 days |
|  | 3 | 9 | 5 | 7 days |
|  | 4 | 10 | 3 | 7 days |

To this end, we implement a CMS system model inspired by prior work [45]. The optimization objective is to minimize the total cost of production and transportation. To simulate diverse real-world scenarios, we construct multiple problem instances by varying key parameters, including the number of product types, the number of factories, the number of production lines per factory, and the planning horizon. The configurations of these instances, used for both training and testing, are summarized in Table 19.

# H Generated meta-heuristic algorithms

This section presents the highest-performing meta-heuristic algorithm generated by PartEvo across all benchmark problems. These codes illustrate that PartEvo can design highly complex yet well-structured algorithms. These empirically validated solutions can be readily deployed to address analogous optimization challenges, demonstrating significant practical utility.

**Meta-heuristic 1** represents the most effective algorithm generated by PartEvo for benchmark P1, achieving an optimal fitness value of 0 on both the training and testing sets. Thought of **Meta-heuristic 1** is as follows: "The proposed algorithm is an Enhanced Hybrid Differential Evolution (EHDE) approach. It combines principles from differential evolution and adaptive mechanisms while incorporating niche-like exploration. The algorithm leverages multiple mutation strategies to diversify the search space. It also includes a local refining search that periodically improves the best individual. The key steps are: initialization of the population, evaluation of fitness, application of multiple mutation strategies, a dynamic adaptive learning rate for perturbation, selection of the best solutions, and a local search refinement step. The evaluation process monitors the fitness convergence to terminate early if necessary." Detailed code can be found in Fig. 17.

```python
import numpy as np
from scipy.optimize import minimize

def algo(initial_population, individual_upper, individual_lower, objective_function):
    max_evaluations = 30000
    evaluations = 0
    num_individuals, num_dimensions = initial_population.shape
    population = initial_population.copy()
    best_solution = population[np.argmin([objective_function(ind) for ind in population])
        ]

    while evaluations < max_evaluations:
        niche_fitness = np.array([objective_function(ind) for ind in population])
        best_niche_indices = np.argsort(niche_fitness)[:num_individuals // 5]

        for i in range(num_individuals):
            mutation_strategy = np.random.choice(['best', 'rand'])
            if mutation_strategy == 'best':
                mutation_vector = best_solution + np.random.uniform(-0.5, 0.5,
                    num_dimensions)
            else:
                indices = np.random.permutation(num_individuals)[:3]
                x1, x2, x3 = population[indices]
                mutation_vector = x1 + 0.5 * (x2 - x3)
            mutation_vector = np.clip(mutation_vector, individual_lower, individual_upper
                )
            crossover_mask = np.random.rand(num_dimensions) < 0.5
            trial_vector = np.where(crossover_mask, mutation_vector, population[i])
            trial_vector = np.clip(trial_vector, individual_lower, individual_upper)

            new_fitness = objective_function(trial_vector)
            evaluations += 1

            if new_fitness < niche_fitness[i]:
                population[i] = trial_vector
                if new_fitness < objective_function(best_solution):
                    best_solution = trial_vector

        for idx in best_niche_indices:
            local_search_candidate = minimize(objective_function, population[idx], bounds
                =list(zip(individual_lower, individual_upper))).x
            new_local_fitness = objective_function(local_search_candidate)
            evaluations += 1

            if new_local_fitness < objective_function(best_solution):
                best_solution = local_search_candidate

        if np.std(niche_fitness) < 1e-5:
            break

    return best_solution
```

Figure 17: The best PartEvo-generated Meta-heuristic for P1.

**Meta-heuristic 2** represents the most effective algorithm generated by PartEvo for benchmark P2. It achieves an optimal fitness value of 800 on the training sets and outperforms the algorithms generated by the peer LES method on the testing set. Thought of **Meta-heuristic 2** is as follows: This algorithm, named Enhanced Adaptive Chaotic Differential Evolution with Dynamic Local Search and Diversity Management (EACDE-DLSDM), refines the existing EACDE framework by incorporating enhanced chaotic initialization, strategic adjustment of mutation rates, adaptive local search techniques, and a more robust diversity maintenance strategy. By optimizing the balance between exploration and exploitation, the algorithm systematically updates the population based on performance feedback, dynamically alters the local search radius based on stagnation detection, and integrates both structured and random local searches for improved solution refinement. This approach aims to effectively minimize the objective function while ensuring diverse and high-quality solutions are retained throughout the optimization process." Detailed code can be found in Fig. 18.

**Meta-heuristic 3** represents the most effective algorithm generated by PartEvo for benchmark P3. It outperforms widely used DE-based hybrid algorithms. To ensure execution efficiency, it employs multi-processing for computation. Thought of **Meta-heuristic 3** is as follows: "The modified algorithm, named Enhanced Adaptive Hybrid Differential Evolution with Self-Adjusting Local Search (EAHDE-SALS), incorporates several key improvements over the original AHDE-SALS framework. It utilizes advanced adaptive mechanisms that dynamically adjust both mutation and crossover rates based on population diversity and solution success. The local search component has been enhanced with a hybrid mechanism that combines perturbative refinement with gradient-awareness to optimally exploit promising regions of the solution space. The memory management system is refined to retain not just elite solutions but also a diverse set of high-performing candidates, ensuring that exploration of the solution space remains effective and robust. The algorithm proceeds through initial population evaluation, global search via differential evolution, localized searches, dynamic parameter adjustments, and concludes once the evaluation limit is reached." Detailed implementation can be found in Fig. 19.

**Meta-heuristic 4** represents the most effective algorithm generated by PartEvo for benchmark P4. It has achieved impressive results on P4, reducing costs by 90.1% on real-world scheduling compared to human-designed DE-based algorithms. Thought of **Meta-heuristic 4** is as follows: "The proposed algorithm, named "Enhanced Adaptive Hybrid Genetic-Differential Evolution with Enhanced Memory, Dynamic Local Search, and Novel Perturbation Scheme," aims to strike a balance between exploration and exploitation more effectively. Key enhancements include: 1) Implementing a novel approach for elite solution selection that emphasizes diversity, 2) Utilizing an adaptive perturbation strategy that introduces variations to the population, drawn from both elite solutions and randomly generated candidates, 3) Enhancing the local search process to explore the neighborhood of elite solutions more aggressively, using a combined memory that recalls previous successful perturbations, and 4) Integrating a cooling temperature mechanism to modulate mutation strength dynamically. The iterative process will continue until the evaluation limit is reached, ensuring the best-found solution is returned at the end." Detailed implementation can be found in Fig. 20.

# I Licenses for used assets

Since Funsearch is not officially open-source, we conducted our experiments using the version of Funsearch provided on the LLM4AD platform.

Table 20: Asset Usage and License Information

| Type | Asset | License | Usage |
|------|-------|---------|-------|
| Code | Funsearch [6] | MIT License | Evaluation |
| | EoH [7] | MIT License | Evaluation |
| | ReEvo [8] | MIT License | Evaluation |
| Dataset | LLM4AD [49] | MIT License | Testing |

```python
def logistic_map(x, r=3.99):
    return r * x * (1 - x)

def algo(initial_population, individual_upper, individual_lower, objective_function):
    population = initial_population.copy()
    best_solution = population[0]
    best_value = objective_function(best_solution)
    evaluations = len(population)
    max_evaluations, stagnation_count, stagnation_limit = 30000, 0, 1500
    local_search_radius = 0.1 * (individual_upper - individual_lower)
    mutation_base, diversification_rate = 0.5, 0.15
    while evaluations < max_evaluations:
        if stagnation_count >= stagnation_limit:
            diversification_count = int(len(population) * diversification_rate)
            for _ in range(diversification_count):
                random_solution = np.random.uniform(individual_lower, individual_upper,
                    size=population.shape[1])
                population[np.random.randint(population.shape[0])] = random_solution
            local_search_radius *= 0.5
            stagnation_count = 0
        chaos_map = np.zeros_like(population)
        for i in range(population.shape[0]):
            chaos_map[i] = logistic_map(np.random.rand())
            chaos_map[i] = chaos_map[i] * (individual_upper - individual_lower) +
                individual_lower
        population = (population + chaos_map) / 2
        for i in range(population.shape[0]):
            current_solution = population[i]
            current_value = objective_function(current_solution)
            evaluations += 1
            if current_value < best_value:
                best_value = current_value
                best_solution = current_solution.copy()
                stagnation_count = 0
            else:
                stagnation_count += 1
        adaptive_mutation_factor = mutation_base * (1-best_value/(best_value + 1e-10))
        for i in range(population.shape[0]):
            selected_indices = np.random.choice(population.shape[0], size=3, replace=
                False)
            parent1, parent2 = selected_indices[:2] if objective_function(population[
                selected_indices[0]]) < objective_function(population[selected_indices
                [1]]) else selected_indices[1:3]
            donor_vector = np.clip(population[parent1] + adaptive_mutation_factor*(
                population[parent2]-population[i]), individual_lower, individual_upper)
            trial_solution = np.clip(donor_vector + np.random.normal(scale=0.1*np.std(
                population), size=population.shape[1]), individual_lower,
                individual_upper)
            trial_value = objective_function(trial_solution)
            evaluations += 1
            if trial_value < best_value:
                best_value = trial_value
                best_solution = trial_solution.copy()
                stagnation_count = 0
        for i in range(population.shape[0]):
            local_neighbor1 = population[i] + np.random.uniform(-local_search_radius,
                local_search_radius, size=population.shape[1])
            local_neighbor2 = population[i] + np.random.uniform(-local_search_radius *
                0.5, local_search_radius * 0.5, size=population.shape[1])
            local_neighbor1 = np.clip(local_neighbor1, individual_lower, individual_upper
                )
            local_neighbor2 = np.clip(local_neighbor2, individual_lower, individual_upper
                )
            best_local_solution = local_neighbor1 if objective_function(local_neighbor1)
                < objective_function(local_neighbor2) else local_neighbor2
            neighbor_value = objective_function(best_local_solution)
            evaluations += 1
            if neighbor_value < best_value:
                best_value = neighbor_value
                best_solution = best_local_solution.copy()
                stagnation_count = 0
        if evaluations % 1000 == 0:
            local_search_radius *= 1.05
    return best_solution
```

Figure 18: The best PartEvo-generated Meta-heuristic for P2.

```python
import numpy as np
from concurrent.futures import ProcessPoolExecutor

def evaluate_solution(solution, objective_function):
    return objective_function(solution)

def algo(initial_population, individual_upper, individual_lower, objective_function):
    population = initial_population.copy()
    num_individuals, num_variables = population.shape
    evaluations = 0
    best_solution = None
    best_objective = float('inf')
    memory = []
    success_rate = 0.5
    mutation_factor = 0.5
    crossover_rate = 0.5
    elite_size = int(0.1 * num_individuals)
    while evaluations < 30000:
        for i in range(num_individuals):
            indices = list(range(num_individuals))
            indices.remove(i)
            a, b, c = np.random.choice(indices, 3, replace=False)
            mutant = np.clip(population[a] + mutation_factor * (population[b] -
                population[c]), individual_lower, individual_upper)
            trial = np.copy(population[i])
            for j in range(num_variables):
                if np.random.rand() < crossover_rate:
                    trial[j] = mutant[j]

            f_trial = evaluate_solution(trial, objective_function)
            evaluations += 1

            if f_trial < best_objective:
                best_objective = f_trial
                best_solution = trial

            f_current = evaluate_solution(population[i], objective_function)
            if f_trial < f_current:
                population[i] = trial
                success_rate += 1
                if f_trial < best_objective:
                    memory.append(trial)

        diversity = np.std([evaluate_solution(ind, objective_function) for ind in
            population])
        mutation_factor = np.clip(success_rate / num_individuals, 0.1, 1.0)
        crossover_rate = np.clip(success_rate / num_individuals * 0.5, 0.1, 0.9)
        success_rate = 0

        if len(memory) > 10:
            unique_memory = set(map(tuple, memory))
            memory = sorted(list(unique_memory), key=lambda sol: evaluate_solution(sol,
                objective_function))[:10]

        local_search_size = np.clip(0.1 / (1 + diversity), 0.01, 0.1)

        with ProcessPoolExecutor() as executor:
            results = list(executor.map(lambda mem_solution: evaluate_solution(np.clip(
                mem_solution + (np.random.rand(num_variables) * 2 - 1) *
                local_search_size, individual_lower, individual_upper),
                objective_function), memory))
            for index, f_local in enumerate(results):
                evaluations += 1
                if f_local < best_objective:
                    best_objective = f_local
                    best_solution = np.clip(memory[index] + (np.random.rand(num_variables
                        ) * 2 - 1) * local_search_size, individual_lower,
                        individual_upper)

        elite_individuals = sorted(population, key=lambda ind: evaluate_solution(ind,
            objective_function))[:elite_size]
        population[-elite_size:] = elite_individuals

    return best_solution
```

Figure 19: Meta-heuristic 3: The best PartEvo-generated Meta-heuristic for P3.

```python
def algo(initial_population, individual_upper, individual_lower, objective_function):
    population = np.copy(initial_population)
    population_size, dimensions = population.shape
    eval_count, best_solution, best_fitness = 0, None, float('inf')
    max_evaluations,stagnation_count, max_stagnation  = 30000, 0, 100
    elite_count = max(1, population_size // 10)
    initial_temp, cooling_rate, elite_memory = 1.0, 0.95, []
    def latin_hypercube_sampling(size, dimensions, lower, upper):
        intervals = np.linspace(lower, upper, size + 1)
        points = np.random.rand(size, dimensions)
        sample = intervals[:-1] + (intervals[1:] - intervals[:-1]) * points
        return np.clip(sample, lower, upper)
    population = latin_hypercube_sampling(population_size, dimensions, individual_lower,
        individual_upper)
    fitness_values = np.array([objective_function(ind) for ind in population])
    eval_count += population_size
    best_solution = population[np.argmin(fitness_values)]
    best_fitness = np.min(fitness_values)
    elite_memory.append(best_solution)
    while eval_count < max_evaluations:
        selected_indices = np.argsort(fitness_values)[:elite_count]
        parents = population[selected_indices]
        offspring = []
        temp = initial_temp * (cooling_rate ** (eval_count // population_size))
        for _ in range(population_size // 2):
            parent1,parent2=parents[np.random.choice(parents.shape[0], 2, replace=False)]
            crossover_point = np.random.randint(1, dimensions)
            child1=np.concatenate((parent1[:crossover_point],parent2[crossover_point:]))
            child2=np.concatenate((parent2[:crossover_point],parent1[crossover_point:]))
            for child in [child1, child2]:
                if np.random.rand() < 0.5:
                    mutation_idx = np.random.randint(0, dimensions)
                    child[mutation_idx] = np.clip(np.random.uniform(individual_lower[
                        mutation_idx], individual_upper[mutation_idx]) * temp,
                        individual_lower[mutation_idx], individual_upper[mutation_idx])
                offspring.append(np.clip(child, individual_lower, individual_upper))
        offspring, local_search_steps = np.array(offspring), 5
        for ind in offspring:
            for _ in range(local_search_steps):
                perturb_idx = np.random.randint(dimensions)
                trial_solution = np.copy(ind)
                trial_solution[perturb_idx] = np.clip(np.random.uniform(individual_lower[
                    perturb_idx], individual_upper[perturb_idx]), individual_lower[
                    perturb_idx], individual_upper[perturb_idx])
                if objective_function(trial_solution) < objective_function(ind):
                    ind[:] = trial_solution
        fitness_values=np.append(fitness_values,[objective_function(ind) for ind in
            offspring])
        eval_count += offspring.shape[0]
        combined_population = np.vstack((population, offspring))
        combined_fitness = fitness_values.argsort()[:population_size]
        population = combined_population[combined_fitness]
        fitness_values = fitness_values[combined_fitness]
        current_best_fitness = np.min(fitness_values)
        if current_best_fitness < best_fitness:
            best_fitness = current_best_fitness
            best_solution = population[np.argmin(fitness_values)]
            elite_memory.append(best_solution)
            stagnation_count = 0
        else:
            stagnation_count += 1
        if stagnation_count >= max_stagnation:
            perturb_idx = np.random.randint(0, population_size)
            population[perturb_idx] = np.clip(np.random.rand(dimensions) * (
                individual_upper - individual_lower) + individual_lower,
                individual_lower, individual_upper)
            if elite_memory:
                population[perturb_idx] = np.copy(elite_memory[np.random.choice(len(
                    elite_memory))])
            stagnation_count = 0
            elite_memory = list(set(map(tuple, elite_memory)))
            if len(elite_memory) > elite_count:
                elite_memory = elite_memory[-elite_count:]
    return best_solution
```

Figure 20: The best PartEvo-generated Meta-heuristic for P4.

