# OpenReview forum: "Partition to Evolve: Niching-enhanced Evolution with LLMs for Automated Algorithm Discovery"
_NeurIPS.cc/2025/Conference — NeurIPS 2025 poster_

### Official Review · Reviewer_S91M · 2025-06-27

**Clarity:** 2
**Significance:** 2
**Originality:** 3
**Rating:** 4
**Confidence:** 1

**Summary:**

The paper proposes "PartEvo", a novel framework integrating niche-based evolutionary computation (EC) techniques with Large Language Model-assisted Evolutionary Search (LES) for Automated Algorithm Discovery (AAD). Recognizing the unique challenges posed by abstract language search spaces in LES, the authors introduce a feature-assisted niche construction method to efficiently partition and explore these spaces. The contributions include a general LES framework, the PartEvo methodology, and comprehensive evaluations demonstrating its superiority over existing human-designed meta-heuristics and other LES methods (e.g., Funsearch and Eoh) on both synthetic and real-world optimization tasks.

**Questions:**

1. I have read the prompts in the appendix and seen the demonstration benchmark functions, but I am still unclear about how these elements work together in practice. Could you please provide a concrete example showing exactly how the prompts are combined with the benchmark tasks in your pipeline?
2. Is it possible to extend this framework towards training a dedicated model, instead of using only a frozen LLM with prompting? For instance, could future work explore jointly fine-tuning or adapting the LLM during the evolutionary search?

**Ethical Concerns:**

["NO or VERY MINOR ethics concerns only"]

**Limitations:**

Yes.

**Paper Formatting Concerns:**

No concerns.

**Quality:**

3

**Strengths And Weaknesses:**

**Strengths**

1. Novel Integration: The paper uniquely integrates niche-based EC methods with LES, addressing effectively the complexity and abstractness of language-based search spaces.
2. Superior Performance: Experimental results consistently demonstrate PartEvo's superior performance across multiple benchmark tasks, significantly outperforming existing human-designed and LES-based methods.
3. Robustness: PartEvo demonstrates strong stability and robustness, achieving consistent high-quality outcomes across varied experimental setups and diverse optimization tasks.

**Weaknesses**

1. The authors could provide more specific examples to better illustrate the proposed method and its advantages clearly.
2. The absence of publicly available resources or clear reproducibility guidelines may limit validation and adoption by the broader research community.

---

> ### Author Rebuttal · Authors · 2025-07-28
>
> Dear Reviewer and Area Chair,
>
> Thank you for your valuable feedback and insightful comments. We genuinely appreciate your review and are pleased to address your questions and suggestions in detail below.
>
> > ### 1.  A Concrete example of how PartEvo works (Question 1 & Weakness 1)
>
> Thank you for requesting a clearer illustration of PartEvo's process. To help readers quickly grasp our methodology, we've now included easy-to-understand pseudocode for PartEvo:
>
> | Line | Description |
> |:-:|:-|
> | | **Input**: Task Description, Training Instances |
> | 1:| Initialize the algorithm population (generated directly by LLM or load existing algorithms) |
> | 2:| Construct algorithm *niches* through feature-assisted niche construction |
> | 3:| &nbsp;&nbsp;&nbsp;&nbsp;**For each $niche$ in *niches*:** |
> | 4:| &nbsp;&nbsp;&nbsp;&nbsp;&nbsp;&nbsp;&nbsp;&nbsp;**For each *operator* in *operators*:** |
> | 5:| &nbsp;&nbsp;&nbsp;&nbsp;&nbsp;&nbsp;&nbsp;&nbsp;&nbsp;&nbsp;&nbsp;&nbsp;Select parent algorithms from *niches* |
> | 6:| &nbsp;&nbsp;&nbsp;&nbsp;&nbsp;&nbsp;&nbsp;&nbsp;&nbsp;&nbsp;&nbsp;&nbsp;Construct prompt (including parent algorithms, task description, and specific instructions related to operator) |
> | 7:| &nbsp;&nbsp;&nbsp;&nbsp;&nbsp;&nbsp;&nbsp;&nbsp;&nbsp;&nbsp;&nbsp;&nbsp;LLM designs offspring algorithm based on the prompt |
> | 8:| &nbsp;&nbsp;&nbsp;&nbsp;&nbsp;&nbsp;&nbsp;&nbsp;&nbsp;&nbsp;&nbsp;&nbsp;Evaluate the offspring algorithm on the training instances |
> | 9 :| &nbsp;&nbsp;&nbsp;&nbsp;&nbsp;&nbsp;&nbsp;&nbsp;&nbsp;&nbsp;&nbsp;&nbsp;Manage the offspring algorithms  based on evaluation results |
> | 10 :| &nbsp;&nbsp;&nbsp;&nbsp;Exit if termination condition is met |
> | 11 :| **Output**: The Designed Algorithm |
>
> To clarify the process, let’s take Benchmark P1 as an example. Our goal here is to discover powerful algorithms that can efficiently solve unimodal optimization problems.
>
> As illustrated in the pseudocode, we start by defining the task (unimodal optimization) and preparing a set of training instances. These instances serve as "practice questions" for PartEvo, enabling us to evaluate whether the generated algorithms are improving. The training instances can be representative unimodal problems or all standard unimodal functions. They provide a consistent way to rank and manage candidate algorithms as they evolve. To assess the generalization ability of the generated algorithms, we use separate training and testing sets. For our experiments, we employ three functions as training instances.
>
> PartEvo continuously optimizes algorithms through an evolutionary process. Each evolutionary loop involves selecting parent algorithms, constructing prompts, generating offspring algorithms, and then evaluating and managing these offspring. Specifically, the generation of offspring algorithms is driven by an LLM, guided by the prompts. The prompt construction follows the template detailed in Appendix C. As PartEvo iterates, the algorithms progressively improve, ultimately leading to high-performing algorithms tailored for unimodal optimization problems.
>
> This process is applied to all our other benchmarks and even real-world applications: simply define the problem and provide relevant training instances, and PartEvo can design a tailored algorithm. We hope this explanation clarifies how PartEvo works in practice.
>
> > ### 2. Extending PartEvo towards training a dedicated model, instead of using only a frozen LLM with prompting (Question 2)
>
> This is a remarkably insightful and forward-thinking suggestion! As mentioned in our paper (lines 579-580), the automated algorithm discovery within PartEvo generates a lot of valuable data on algorithm design, including step-by-step creation processes and various solution attempts for specific problems. This rich data can be used offline to fine-tune LLMs, further enhancing their capabilities in algorithm design.
>
> An exciting future direction is the concept of training the LLM "on the fly" as the evolutionary search progresses. Recent research, such as [1], has begun exploring this idea, and we also plan to investigate this in our future research. We believe this approach could enable us to create a smaller, specialized language model that, even with a reduced scale, could achieve powerful algorithm discovery abilities when combined with our evolutionary framework. This would be a significant step toward reducing the computational demands for LES-based algorithm design, making it more accessible and valuable to the wider community.
>
> [1] Huang, Ziyao, et al. "Calm: Co-evolution of algorithms and language model for automatic heuristic design." arXiv preprint arXiv:2505.12285 (2025).
>
> > ### 3. Publicly available resources （Weaknesses 2）
>
> Thank you for this valuable suggestion. We fully agree that making our work accessible is essential. We have already organized PartEvo's code, along with the benchmarks and scripts needed to reproduce our comparison experiments.
>
> We plan to make our GitHub repository public as soon as the NeurIPS decisions are announced. This will enable other researchers to easily verify our findings, gain a detailed understanding of our methods, and build upon our work. Additionally, we intend to expand our experiments to cover a wider range of problems, providing more comprehensive insights. We genuinely hope that our work inspires further research and serves as a valuable resource for the community.
>
> ---
>
> In summary, we deeply appreciate the reviewer's constructive feedback, which has helped us significantly enhance the clarity and impact of our paper. We are committed to incorporating all the revisions and clarifications detailed above into our final revised version.

---

> > ### Comment · Reviewer_S91M · 2025-08-05
> >
> > Thank the authors for the rebuttal. You have addressed most of my concerns.

---

### Official Review · Reviewer_DVie · 2025-07-02

**Clarity:** 3
**Significance:** 3
**Originality:** 3
**Rating:** 5
**Confidence:** 4

**Summary:**

This paper presents PartEvo, a novel Language-based Evolutionary Search (LES) method that improves algorithm discovery efficiency. It introduces abstract search space partitioning to enable structured niche construction for algorithm discovery, integrating evolutionary computation techniques to enhance sampling resource allocation. PartEvo combines verbal gradients with local and global search strategies to discover high-performing algorithms, demonstrating significant efficiency gains through the fusion of evolutionary computation with LES. The method shows superior performance on algorithm automatic discovery tasks across both synthetic and real-world optimization problems, outperforming human-designed meta-heuristic baselines and proving the value of feature-assisted niche construction, advanced prompting, and niche-based sampling. PartEvo particularly excels in scenarios with limited sampling budgets by efficiently partitioning language search spaces and constructing niches that enable more effective allocation of LLMs queries.

**Questions:**

- ReEvo is also mentioned as an AAD approach. Why wasn't it included as a baseline in your experiment?
- How does PartEvo perform regarding time and cost/token consumption?
- Line 267, 280: "EoH" vs. "Eoh"—please be consistent.

**Ethical Concerns:**

["NO or VERY MINOR ethics concerns only"]

**Final Justification:**

Rebuttal of the authors mainly addressed my concerns, and I decide to keep the original positive score.

**Limitations:**

Yes, in the appendix

**Quality:**

3

**Strengths And Weaknesses:**

**Strengthen**

- Propose an approach integrating niche-based EC techniques that largely improve algorithm automatic discovery efficiency.
- Comprehensive experiment results demonstrate that it out-performs existing baselines.

**Weakness**

- It would be helpful if the author could include experimental results on classical optimization problems such as TSP, job scheduling, etc.
- Evaluation with LLMs of various capabilities would be necessary to determine how sensitive the approach is to different LLMs.

---

> ### Author Rebuttal · Authors · 2025-07-29
>
> Dear Reviewer and Area Chair,
>
> Thank you for your comprehensive review and valuable feedback. We appreciate your acknowledgment of the contribution our PartEvo framework makes to automatic algorithm discovery. Your insights have been instrumental in enhancing our paper. Below, we provide our responses to your questions and suggestions.
>
> > ## Question 1: ReEvo as a baseline
>
> We sincerely appreciate the Reviewer's suggestion to include ReEvo as a baseline. As recommended, we have integrated ReEvo as a primary baseline for comparison in the main experiments for benchmarks P1-P4. While ReEvo was previously used for the Online Bin Packing (OBP) problem in Appendix B.4, its inclusion in the core experiments enhances the rigor of our study, providing a more comprehensive evaluation of PartEvo's capabilities.
>
> To ensure a fair comparison, we use the same initial population for ReEvo as we did for other LES methods and run it with the same sample budget to discover the algorithms for benchmark P1-P4. The average results from three independent runs are presented in the table below (an expanded version of the original Table 8):
>
> | Problem | Instances | Funsearch | EoH    | ReEvo  | PartEvo  |
> |-|-|-|-|-|-|
> | **P1**  | Training  | 9.769     | 1.846  | 1.572  | **0.074** |
> |         | Testing   | 0.879     | 1.646  | 1.013  | **0.023** |
> | **P2**  | Training  | 800.09    | 800.98 | 800.00 | **800.00**|
> |         | Testing   | 439.31    | 368.95 | **364.19** | 365.20  |
> | **P3**  | Training  | 7042.21   | 7268.22| 6564.44| **6539.24** |
> |         | Testing   | 57696.53  | 61142.93| 57455.34 | **56753.77** |
> | **P4**  | Training  | 24289.3   | 19152.4| 11102.3 | **6424.7**  |
> |         | Testing   | 1.67${\times}10^5$ | 77134.3 | 79795.2 | **62731.4** |
>
> Consistent with our previous findings in the OBP problem (lines 621-624), both PartEvo and ReEvo consistently outperform EoH and Funsearch across all benchmarks. This highlights the benefits of leveraging the "verbal gradient" concept, specifically through the application of "Reflection" and "Summary" for enhanced algorithm discovery. Importantly, thanks to PartEvo's unique EC-inspired operators (such as CN and LGE), it outperforms ReEvo. This demonstrates that both "verbal gradient" and advanced EC techniques can enhance search efficiency and should be equally emphasized in future LES research.
>
>
> > ## Question 2: Time and cost/token consumption
>
> In terms of time consumption, the total time required for PartEvo to design algorithms can be broadly divided into two main components: LLM-driven algorithm generation and the evaluation of the generated algorithms. Typically, for a given number of queries, the evaluation time constitutes the majority of the total time, especially as the complexity of problem evaluation increases. For example, with a budget of 500 samples, the P1 benchmark, which involves three 20-dimensional standard test functions, requires approximately 1 hour. In contrast, the P3 benchmark, which includes two real-world scheduling instances, requires 2 to 3 hours. It is important to note that this still represents a significant efficiency gain compared to human experts, and the system can operate tirelessly.
>
> Cost consumption is directly related to token usage and the specific LLM employed. Compared to EoH, PartEvo's "Reflection" and "Summary" operations (e.g., RE and SE operators) lead to greater token consumption than simpler crossover and mutation operations. For instance, the SE operator requires including the "thoughts" of all algorithms stored in the Sorted External Archive as part of the prompt for summarization by the LLM. When we set the archive size to 40, token consumption increases by 4 to 5 times compared to simpler operations like Crossover. (e.g., it requires about 1,500 tokens with the Crossover operator vs. 7,500 tokens with the SE operator).
>
> Overall, for real-world problems such as P3 and P4, a single run of PartEvo with 500 samples using GPT-4o-mini costs approximately 0.14 USD, which we consider acceptable. Furthermore, with anticipated advancements in computational power, this cost is expected to decrease in the future.
>
> > ## Question 3: Consistency in writing of EoH
>
> Thank you for your careful observation. We will ensure consistency with the notation used in the EoH paper. We have corrected this issue and ensured that all relevant sections of the paper are now consistent.
>
> > ## Suggestion 1: Include experimental results on classical optimization problems
>
> We sincerely appreciate this valuable suggestion. We agree that expanding our experimental results to cover a broader range of classical optimization problems would better demonstrate the universality of the PartEvo framework. We intend to include these problems in our future experiments and will share the findings on our open-source website soon. We believe that these additional experiments will further validate the applicability and advantages of the PartEvo, thereby boosting the community's confidence in the use of EC techniques within the LES paradigm.
>
> > ## Suggestion 2: Evaluation with LLMs of various capabilities
>
> Thank you for your suggestion. We also recognize the impact of LLM capabilities on the overall algorithm discovery performance of PartEvo, as discussed in Appendix A.3. In response, we have added experiments using GPT-4o and GPT-3.5-turbo on the P4 benchmark, with an identical initial population, a 500-sample budget, and three independent runs. To improve readability, we have sorted the results of each model's three runs by performance for display. Below are the results for each run, the average, and the Standard Error of the Mean (SEM):
>
> | LLM  | Run 1  | Run 2  | Run 3  | Average | SEM |
> |:------------:|:------:|:------:|:------:|:-------:|:--------------:|
> | GPT-3.5-turbo      |  4924.5 |  9200.2 | 13261.9 |  9128.9 |       1965.360 |
> | GPT-4o-mini  |  4539.1 |  4679.5 | 10055.4 |  6424.7 |       1482.610 |
> | GPT-4o       |  3304.6 |  6028.0 |  8187.5 |  5840.0 |       1153.466 |
>
>
> As shown in the table, we observe that the higher the LLM's capability, the better and more stable the performance of PartEvo. As mentioned in lines 541-545, as LLMs continue to evolve, we anticipate improvements in PartEvo's algorithm discovery capabilities.
>
> ---
>
> Once again, we sincerely thank the Reviewer for acknowledging our work and offering such valuable guidance. With these improvements, we believe our paper will be more widely recognized.

---

### Official Review · Reviewer_h8jf · 2025-07-02

**Clarity:** 2
**Significance:** 2
**Originality:** 3
**Rating:** 3
**Confidence:** 3

**Summary:**

The paper proposes an LLM-assisted evolutionary search (LES) framework for Automated Algorithm Discovery (AAD) that incorporates niching-based collaborative search in the LES abstract language space. The proposed approach (PartEvo) combines verbal gradients with search strategies, leading to high performing algorithms for the AAD task. The method is evaluated on both synthetic and real data, showing its advantage over human-designed metaheuristic algorithms.

**Questions:**

1. Why did you consider k-Means algorithm and not other clustering methods?
2. How internal parameters of PartEvo were selected? How sensitive is the method to their choice?
3. Could you compare your approach with the one proposed in [1] (see Weaknesses)?

**Ethical Concerns:**

["NO or VERY MINOR ethics concerns only"]

**Final Justification:**

My concerns are mostly addressed. Still, since the paper does not provide strong theoretical analysis, I would expect more thorough experimental evaluation. Also, the question of how to choose the algorithm’s parameters is not fully addressed. I’m raising my score to 3: Borderline reject.

**Limitations:**

Yes

**Paper Formatting Concerns:**

There are no formatting concerns.

**Quality:**

2

**Strengths And Weaknesses:**

Strengths:
1. An interesting extension of LES algorithm through incorporation of EC-niching method.
2. In experimental evaluation on 4 benchmark problems the methods performs comparably or better than the considered baselines.


Weaknesses:
1. The proposed approach lacks deeper investigation of parameter settings. For instance, what is the relation between $k$ and $m$, and how should their values be selected for a given problem at hand?
2. The experimental evaluation is not sufficient. More diverse benchmarks (especially related to real-life problems) should be considered for a thorough assessment of PartEvo. Furthermore, mean and standard deviation results should be presented within Table 1 in the main paper.
3. Besides standard population-based metaheuristics, PartEvo is evaluated only against two specialised methods, which does not allow for making definitive conclusions regarding its strengths and weaknesses. Other methods, e.g. [1], should be considered in the experimental assessment of the method.
4. The contribution of the paper is not precisely stated. To my understanding it is an extension of LES approach by means of adding a niching algorithm, however the idea of niching in EC is clearly not novel. I understand that the novelty lies in the application of niching to a specific search space, however, this space specificity and the consequent difficulty of designing the niching method is not clearly explained in the paper.

[1] ReEvo: Large Language Models as Hyper-Heuristics with Reflective Evolution, [NeurIPS 2024]

---

> ### Author Rebuttal · Authors · 2025-07-29
>
> Dear Reviewer and Area Chairs,
>
> Thank you for your thorough review and constructive suggestions. We appreciate your insights and have carefully addressed all the concerns raised in the following five key points.
>
> > ## Q1：Why K-Means? Can PartEvo be compatible with other clustering methods? (Question 1)
>
> In PartEvo, clustering serves as a tool for grouping individuals within the feature space. As we mentioned in lines 136-137, PartEvo is compatible with various clustering methods.
>
> We specifically chose K-Means due to its simplicity and transparency, as noted in lines 169-170. This deliberate choice allows us to isolate and confirm that the observed performance improvements are primarily due to the effects of niche-based evolution, rather than influenced by the complexities of more advanced clustering methods.
>
> To further demonstrate PartEvo's flexibility and compatibility, we conduct additional experiments considering three clustering methods: K-means, Gaussian Mixture Model (GMM), and Spectral Clustering (SC). Each version of PartEvo used the same feature mapping and initial population, with three independent runs per method. The experimental results are presented in the table below:
>
> |Methods|P1| P2 |P3 | P4 |
> |:-|:-:|:-:|:-:|:-:|
> |EoH | 1.846 | 800.98 | 7268.22 |19152.4|
> |PartEvo-K-means| 0.074 | 800.42 | 6539.24 | 6424.7|
> |PartEvo-GMM | 0.048 | 800.10 | 6580.82 | 5423.5 |
> |PartEvo-SC | 0.075 | 800.01 | 6543.54 |6714.2 |
>
> As shown in the table, PartEvo consistently outperforms EoH across all benchmarks, regardless of the clustering method employed. These results strongly support our claims that PartEvo is compatible with various clustering algorithms.
>
> In summary, (1) K-Means was chosen for its simplicity, transparency, and broad familiarity within the research community; (2) PartEvo can seamlessly integrate with other clustering methods, as demonstrated by the results presented above.
>
> > ## Q2: Internal Parameter Selection and Sensitivity (Question 2 and W1)
>
> We appreciate your inquiry about the internal parameter settings and their sensitivity.
>
> For parameter $m$, we adopted the settings established in the EoH paper to ensure a consistent baseline. As for parameter $k$, which determines the number of parents selected from other niches and influences the length of the prompts given to the LLMs, we set it to 2 based on empirical insights and practical considerations.  This choice helps avoid excessively long prompts, as longer inputs can reduce the performance of LLMs and potentially lead to hallucinations, as noted by Levy et al.[1].
>
> Furthermore, we conducted a detailed investigation into the effect of varying $K$ value under a fixed population size, as outlined in lines 313-332. The value of $K$ is crucial for balancing exploration and exploitation within the search process. As shown in Table 3, the trends for $K$ are consistent across both easy and hard problems. This suggests a practical approach: $K$ can be effectively tuned on simpler problems (where computational costs are lower) and then applied to more complex tasks, thereby enhancing the applicability of PartEvo.
>
> Given the exploratory nature of this paper at the framework level, we have demonstrated the viability of the niche-enhanced framework. However, we acknowledge that a comprehensive analysis of the interactions between parameters has not yet been conducted at this stage. Despite this, our current parameter settings have yielded promising results on problems P1-P4 and the Online Bin Packing problem, demonstrating their applicability in supporting algorithm design for various tasks. Moving forward, we are committed to extending this analysis and providing a more comprehensive evaluation on a publicly accessible platform. Thank you for your constructive suggestion!
>
> [1] Levy, Mosh, Alon Jacoby, and Yoav Goldberg. "Same task, more tokens: the impact of input length on the reasoning performance of large language models." arXiv preprint arXiv:2402.14848 (2024).
>
> > ## Q3: Concern about the sufficiency of experimental evaluation (Weakness 2)
>
> We would like to clarify that we have used **five** distinct benchmarks with varying levels of difficulty to evaluate PartEvo, which we believe provides a robust and comprehensive evaluation.
>
> Specifically, P1 and P2 are standard unimodal and multimodal test functions widely used in optimization research. **P3 and P4 represent more complex, real-world industrial scheduling problems**. Additionally, we conducted extensive experiments on the Online Bin Packing benchmark, which is detailed in Appendix B.4, further broadening the scope of our evaluation.
>
> Regarding the mean and standard deviation results, they are presented in Appendix B.2. We understand that greater accessibility is desirable, and in response to your suggestion, we will incorporate these detailed statistical results into the main paper in the revised version, making them more readily available to readers.
>
> > ## Q4: Comparison with ReEvo (Question 3 and W3)
>
> We highly value the reviewer's suggestion to compare PartEvo with ReEvo. In response, we have included ReEvo as one of our primary experimental baselines for comparison. While ReEvo was previously incorporated as a baseline comparison for the Online Bin Packing (OBP) problem in Appendix B.4, its integration in the main experiments for P1-P4 significantly enhances the rigor of our study, offering a more comprehensive evaluation of PartEvo's capabilities.
>
> For a fair comparison, we use the same initial population for ReEvo as for other LES methods and run it with the same sample budget to discover the algorithms for benchmark P1-P4. The average results from three independent runs are presented below (this is an expanded version of the original Table 8):
>
> | Problem | Instances | Funsearch | EoH    | ReEvo  | PartEvo  |
> |-|-|-|-|-|-|
> | **P1**  | Training  | 9.769     | 1.846  | 1.572  | **0.074** |
> |         | Testing   | 0.879     | 1.646  | 1.013  | **0.023** |
> | **P2**  | Training  | 800.09    | 800.98 | 800.00 | **800.00**|
> |         | Testing   | 439.31    | 368.95 | **364.19** | 365.20  |
> | **P3**  | Training  | 7042.21   | 7268.22| 6564.44| **6539.24** |
> |         | Testing   | 57696.53  | 61142.93| 57455.34 | **56753.77** |
> | **P4**  | Training  | 24289.3   | 19152.4| 11102.3 | **6424.7**  |
> |         | Testing   | 1.67${\times}10^5$ | 77134.3 | 79795.2 | **62731.4** |
>
> Consistent with our previous findings in the OBP problem (lines 621-624), both PartEvo and ReEvo consistently outperform EoH and Funsearch across all benchmarks. This highlights the benefits of leveraging the "verbal gradient" concept, specifically by applying "Reflection" and "Summary" for enhanced algorithm discovery. Importantly, thanks to PartEvo's unique EC-inspired operators (such as CN and LGE), it outperforms ReEvo. This demonstrates that both "verbal gradient" and advanced EC techniques can enhance search efficiency and should be equally emphasized in future LES research.
>
> > ## Q5: Further Clarification on Paper's Contribution(Weakness 4)
>
> As noted by other reviewers, we have indeed made efforts to clearly state our contributions. Here, we are pleased to offer further clarification for your consideration.
>
> Our primary aim is to introduce the niching as an example to demonstrate that established EC techniques can drive more sustainable progress in the LES paradigm. Importantly, our work goes beyond simply applying niching. We uncover the challenges faced when applying it, particularly when the search space is extended beyond traditional numerical or manually designed discrete spaces into abstract language spaces. To address this, we introduce feature-assisted niche construction to support the implementation of niching techniques within the LES framework. The experimental results clearly show that niching enhances the performance, efficiency, and stability of the LES framework. We hope that our work will encourage the research community to recognize that techniques refined over decades in EC can significantly contribute to the evolution of this emerging LES paradigm.
>
> For your reference, we briefly discuss the abstract nature of the LES search space in lines 36-42. Furthermore, we explicitly describe the space specificity and the consequent difficulty of introducing niching in lines 101-106, with our proposed solution outlined in lines 107-110. We hope this clarification helps in understanding the contributions of our work.
>
> ---
>
> In summary, we hope that PartEvo serves as a significant step in bridging mature EC techniques with the emerging LES paradigm, offering valuable insights for the research community. We sincerely appreciate your constructive feedback and the time you dedicated to our work. We hope these responses address your concerns, and we look forward to your reconsideration of the rating.

---

### Official Review · Reviewer_pyQe · 2025-07-03

**Clarity:** 3
**Significance:** 2
**Originality:** 2
**Rating:** 3
**Confidence:** 5

**Summary:**

This paper discuss a popular topic in recent optimization community: using large language models for automated optimization algorithm design. The authors propose a refined procedure termed as PartEvo to improve existing works such as EoH and Funsearch. Inspired by the success of niching strategy in traditional evolutionary algorithms, the authors first propose adding niching-based searching process into EoH framework. To this end, the authors first address the challenge of measuring similarity in abstract program space, where two metrics are proposed: code similarity and thought similarity. The former is based on the BLEU score of the generated algorithm sourcecodes and the latter is based on the hidden feature of the thought text (as those in EoH) through a pretrained BERT model. Given the similarity metric, the authors further add two additional searching strategies into the algorithm evolution process, which provide effective program search within similar algorithm individuals and balance the exploration and exploitation tradeoff in the same time. The experimental results, including the comparison test and ablation study, demonstrates this work shows better performance than EoH and Funsearch, and its proposed niching-related technique is effective.

**Questions:**

See Strengths And Weaknesses above.

**Ethical Concerns:**

["NO or VERY MINOR ethics concerns only"]

**Final Justification:**

I am maintaining my original recommendation that this paper is not ready for acceptance at NeurIPS. While I appreciate the authors' detailed, multi-stage rebuttal, the revisions and arguments presented do not sufficiently address the core weaknesses of the manuscript.

First, as I indicated, the primary contribution of this work—integrating a niching technique into the LES framework—is a useful incremental improvement. However, it does not strike me as a groundbreaking or sufficiently novel contribution to meet the high standards of this venue. Given the limited theoretical novelty, the burden of proof falls heavily on the experimental section to demonstrate significant and insightful results. Unfortunately, the experimental validation remains a weak part of this paper.

**The new experiment hastily added to compare against AM, POMO, and LEHD is problematic.** First, the major experiments conducted in the paper  focused on numeric optimization, yet the authors chose to compare against methods that are primarily designed for combinatorial problems like the TSP, seemingly for the sake of a quick comparison, rather than selecting from the many existing learning-for-optimization algorithms relevant to their domain. Second, I have strong reservations about the conclusion that LES can outperform a strong baseline like LEHD. This claim is likely biased, as it is based on tests conducted on a very small and insufficient sample of only six instances.

Besides, **the lack of standardized benchmarking problems persisted.** The paper's numerical experiments are still not evaluated on any widely-used, standard benchmark test sets, such as the COCO or CEC suites. Instead, the authors rely on their own newly constructed dataset, which makes it difficult to compare the method's performance and position it within the broader field.

**Limitations:**

I strongly suggest the authors discuss the limitations of this paper. Using LLMs for algorithm design is promising. However, the true potential, the efficiency and the generalization boundary are very aspects that deserve in-depth analysis.

**Quality:**

3

**Strengths And Weaknesses:**

Pros:

1. The writing of this paper is generally clear.
2. The motivation of this paper is properly backed up.
3. The methodology, including the similarity metrics and the corresponding niching-based searching enhancement, is well-organized and technically rational.
4. Considering existing LLM-assisted optimization framework, this work presents certain novelty and performance improvement. The potential audience of this work is hence moderate.

Cons:

1. Overall, I think the contribution of this paper is incremental. Applying evolution-like thoughts for algorithm design is novel, such as EoH. Employing high-level evolutionary technique such niching, however, is incremental.
2. I still think the authors overpromise the significance of this work, since they claim that “Importantly, the proposed framework is highly flexible and extensible.” (line 134), and “ Additionally, any feature and clustering method can be utilized” (lines 136-137). For a framework, at least two or more scenarios should be validated. I suggest the authors add more experimental analysis to validate the general usage of the proposed framework.
3. There might be an unclear methodology issue. Given the two similarity metrics the authors have proposed, how to use them in the proposed framework? I can not find any detail of how to combine these two metrics, this further lead to my confusion on the experimental results in Table 1, are the results of PartEvo obtained by aggregating the results of PartEvo with CSV and PartEvo with TE? If so, the robustness of PartEvo would be highly dependent on the metric selection, which is somewhat not so ideal.
4. Although in experiments, four optimization scenarios are validated, the experimental setting is still too simple. For example, in P1 and P2, the training instances and testing instances are “small size with simple structure (sphere function)”, which makes me doubt the generalization ability of the learned algorithm. Adding more diversity in your testset to validate the true potential of the generated optimizer.
5. The experiments still lack broader comparison with existing automated algorithm design methods such as those using Genetic Programming and Reinforcement Learning. The training efficiency, the resource requirement and the final performance should be fairly discussed.
6. Review on the related works is quite limited. For “Automated algorithm discovery” part, many valuable papers should be recognized and carefully reviewed to respect the scientific integrity and historical evolution path in this field, such as
van Stein, N., & Bäck, T. (2024). Llamea: A large language model evolutionary algorithm for automatically generating metaheuristics. IEEE Transactions on Evolutionary Computation.
Yu, H., & Liu, J. (2024). Deep Insights into Automated Optimization with Large Language Models and Evolutionary Algorithms. arXiv preprint arXiv:2410.20848.
Ma, Z., Guo, H., Gong, Y. J., Zhang, J., & Tan, K. C. (2025). Toward automated algorithm design: A survey and practical guide to meta-black-box-optimization. IEEE Transactions on Evolutionary Computation.

---

> ### Author Rebuttal · Authors · 2025-07-28
>
> Dear Reviewer and Area Chairs:
>
> Thank you for your thoughtful review and constructive suggestions. We appreciate your insights and respond to each of your concerns in detail below.
>
> > ### Q1: The contribution of this paper seems incremental?
>
> We would like to clarify the significance of our work. Integrating large language models (LLMs) with evolutionary computation (EC) for algorithm design is still an emerging research field. While some initial methods have explored the basic combination of the two using simple evolutionary mechanisms such as crossover and mutation, EC itself has decades of research focused on advanced mechanisms designed to improve performance, efficiency, generalization, and more. These mature EC techniques hold the potential for driving progress in this emerging LES paradigm. In this context, we introduce **niching** as an example to demonstrate how EC techniques can contribute to more sustainable progress.
>
> Importantly, our work goes beyond simply applying niching. We uncover the challenges faced when applying it, particularly when the search space is extended to the implicitly shaped language space. To address this, we introduce *feature-assisted niche construction* to support the implementation of niching techniques within the LES framework. Our experimental results show that niching significantly enhances the performance, efficiency, and stability of the LES framework.
>
> Our work is expected to deepen the community's understanding of integrating LLMs with EC and inspire further innovative thinking. Although our contribution may seem incremental, it is a necessary step that lays the groundwork for more significant advancements in this field. We are confident that these improvements will lead to profound breakthroughs in the future.
>
> > ### Q2: Overpromising the flexibility and general validation of the PartEvo?
>
> We would like to clarify our claim of the framework's flexibility and extensibility. When we state "flexible and extensible (line 134)," we mean that our niching-enhanced framework is compatible with the core concepts from works like EoH and ReEvo (e.g., PartEvo incorporates concepts such as "Thought" from EoH and "verbal gradients" from ReEvo). As future advancements are made, the niching-enhanced framework can continue to evolve and integrate new techniques.
>
> Regarding the general validation concern:
>
> (1) Features:  We have proposed two feasible features (CSV and TE) to validate our framework, demonstrating their value compared to random features, as shown in **Table 4**.
>
> (2) Clustering Method: To intuitively demonstrate PartEvo's compatibility with various clustering methods, we conduct additional experiments considering three clustering approaches: K-means, Gaussian Mixture Model (GMM), and Spectral Clustering (SC). In this experiment, each version of PartEvo uses CSV features and the same initial population, with three independent runs for each method. The average results are presented below:
>
> |Methods|P1| P2 |P3 | P4 |
> |:-|:-:|:-:|:-:|:-:|
> |EoH | 1.846 | 800.98 | 7268.22 |19152.4|
> |PartEvo-K-means| 0.074 | 800.42 | 6539.24 | 6424.7|
> |PartEvo-GMM | 0.048 | 800.10 | 6580.82 | 5423.5 |
> |PartEvo-SC | 0.075 | 800.01 | 6543.54 |6714.2 |
>
> As the table illustrates, PartEvo, regardless of the clustering method employed, consistently outperformed EoH across all tested scenarios. These results strongly confirm our claims regarding the framework's compatibility with various clustering methods.
>
> These experimental validations indeed demonstrate that our framework is flexible and extensible, supporting our original claims.
>
> > ### Q3: Unclear statement regarding the use of two similarity metrics.
>
> We sincerely apologize for any confusion caused by the unclear explanation. To clarify, all results presented in the main text are based on PartEvo using **CSV** as the feature, except for Table 4, which examines the impact of different features. We will provide a more explicit clarification of this in the Implementation Details section.
>
> Thank you for bringing this to our attention. We hope this resolves any confusion. Additionally, as shown in **Appendix B.5** (lines 625-639), both CSV and TE versions of PartEvo exhibit excellent robustness.
>
> > ### Q4: Concerns about the benchmark setting and generalization ability of the learned algorithm.
>
> We appreciate the reviewer’s concerns and would like to clarify our benchmarking efforts. We designed benchmarks with varying levels of difficulty to thoroughly evaluate PartEvo. Benchmarks P1 and P2 involve standard unimodal and multimodal test functions that are commonly used in optimization research, while P3 and P4 introduce more complex, real-world industrial optimization problems.
>
> As detailed in **Appendix F**, we set different training and testing instances for each benchmark. For example, in P1 (see Table 13), the training instances consist of simple sphere functions, whereas the testing instances include more complex functions such as the logistic and Gaussian functions that are outside the training data. To further increase the diversity of the testing set, we also introduced shift terms. Additionally, to better assess the scalability of the learned algorithm, we incorporated a variety of complex test instances in P3 and P4. As shown in **Table 15**, these instances vary in terms of dimensions, scenarios, and optimization complexity.
>
> It is important to note that **Table 1** shows that PartEvo’s algorithms perform well on the testing instances, indicating a certain level of generalization ability. We hope this clarification addresses your concerns. Furthermore, we fully agree with the reviewer on the importance of generalization and provide a more detailed response to this issue in **Q7**.
>
> > ### Q5: Broader discussion about comparisons between LES and Genetic Programming (GP) or Reinforcement Learning (RL).
>
> Thank you to the reviewer for the insightful suggestion. Our main contribution lies in integrating EC techniques to enhance the LES paradigm, which is why our experiments focused on comparisons with existing LES methods. However, we welcome further discussion regarding the comparisons between LES and GP or RL.
>
> Reference [1] provides a detailed comparison between GP and LES, showing that GP generally saves time and requires fewer resources in a single code generation step, as it does not involve interactions with LLMs. However, LLMs help overcome structural limitations and leverage external knowledge, improving code quality in a generation step and reducing the total number of generation steps, thereby accelerating problem-solving. Recent studies [2],[3] also emphasize that LLMs enhance the flexibility and intelligence of methods, decreasing reliance on human experts and improving performance, which goes beyond traditional resource concerns. Regarding RL, there are currently no known studies providing a direct comparison between RL and LES. However, we believe the LES paradigm's advantage lies in avoiding the need to retrain neural networks for each task, along with an interpretable algorithm design process.
>
> Overall, we agree that rethinking the relationship between GP, RL, and LES is valuable and merits future exploration. Our intention is not to compare them in a competitive manner but rather to acknowledge the significant potential of new tools like LLMs. These tools hold great promise and require further research, such as this paper, to optimize their utilization.
>
> [1]Hemberg, Erik, et al. "Evolving code with a large language model." Genetic Programming and Evolvable Machines 25.2 (2024): 21.
>
> [2]Zheng, Kefeng, et al. "CST-LLM: Enhancing Airfoil Parameterization Method with Large Language Model." Aerospace Science and Technology (2025): 110548.
>
> [3]Liu, Max, et al. "Synthesizing programmatic reinforcement learning policies with large language model guided search." arXiv preprint arXiv:2405.16450 (2024).
>
> > ### Q6: Suggestions for expanding the review of related works in automated algorithm discovery.
>
> Thank you for your insightful feedback. We fully agree that a more comprehensive and up-to-date review of the rapidly evolving field of automated algorithm discovery is essential. We will make sure to include and carefully examine the works you mentioned, along with other key references. This will provide a broader context and enable us to better position our approach within the existing literature.
>
> Once again, thank you for your constructive suggestions. We are confident that these revisions will significantly strengthen the manuscript.
>
> > ### Q7: Expanding the discussion on limitations, including generalization boundaries.
>
> Thank you for your constructive suggestion. We would like to clarify that we have already discussed several key limitations in the "Discussion and Limitations" section of Appendix A (lines 540-574), including the efficiency aspects in lines 562-574. We are also happy to further discuss other aspects, such as generalization boundaries.
>
> In the LES paradigm, the automatic discovery process of algorithms relies on evaluations over training instances. Similar to supervised learning, training on fixed data may lead to overfitting. Currently, the algorithms learned by PartEvo perform well within the distribution of training instances but may not generalize equally well to instances that are outside this distribution. We recognize this limitation and are actively researching multi-distribution cooperative evolution to improve generalization. Additionally, many techniques developed in machine learning to address generalization issues can be adapted and applied within the LES paradigm.
>
> We will include a more detailed discussion of these aspects in the revised manuscript. Once again, thank you for your insightful and professional suggestions. We believe that these clarifications will enhance the manuscript and provide a more comprehensive understanding of the work's limitations and future directions.

---

> ### Author Response · Authors · 2025-08-09
>
> Dear Reviewer,
>
> Thank you for your valuable feedback. We have carefully considered your latest comments and have included a detailed discussion comparing the LES paradigm with deep learning-based AAD methods, specifically addressing training efficiency, resource requirements, and performance.
>
> We sincerely hope this **new evidence** will be taken into consideration for your final evaluation. Thank you again for your time and invaluable guidance.
>
> Sincerely,
>
> The Authors

---

### Author Response · Authors · 2025-08-09

Dear Reviewers and Area Chairs,

We sincerely thank all of the reviewers and area chairs for your careful reading of our manuscript, your constructive feedback, and the time you dedicated to the review process.

During the rebuttal period, we made every effort to address all reviewer comments through substantial additional work, which includes:
- **Expanding our baseline comparison** by including ReEvo for a more comprehensive evaluation.
- **Validating PartEvo’s compatibility** with various clustering methods.
- **Providing a detailed discussion on algorithm parameter selection** and performing sensitivity analysis to better understand the impact of different choices.
- **Investigating the influence of LLM capabilities** on overall algorithm discovery performance.
- **Comparing the LES paradigm with Genetic Programming and Deep Learning-based** algorithm design methods in terms of training efficiency, resource requirements, and performance.
- **Adding specific examples** to effectively illustrate PartEvo and highlight its advantages.

Beyond these additional experiments, our contribution represents **the first comprehensive attempt** to address a fundamental challenge in the LES paradigm: how to integrate advanced EC techniques, like Niching, given the **significant shift in the search space** from structured numerical/genetic representations to abstract language space. While much recent work focuses solely on prompt engineering with simple evolutionary operators, our approach is distinct. We demonstrate that both prompt engineering and advanced EC techniques can jointly enhance search efficiency. We believe this dual focus is crucial for future LES research and points the way for incorporating decades of EC wisdom into this promising interdisciplinary field, thereby fostering more sustainable progress.

We also note that the NeurIPS review guidelines recognize both algorithmic and framework-level contributions as valuable, provided they meaningfully advance the state of the art. Our work aligns with this standard by offering a critical technical foundation and methodological paradigm for integrating a broader range of advanced EC techniques (e.g., diversity maintenance, multi-objective optimization, adaptive evolutionary mechanisms) into LES. This marks a significant step forward in exploring the full potential of complex EC techniques within the LES paradigm.

We have carried out these experiments, clarified methodological details, and refined explanations right up to the final day of the rebuttal period. We have never avoided any of the reviewers’ concerns or questions — we have addressed each one directly and in full. We sincerely hope that the completeness and transparency of our responses convey our commitment to rigorous evaluation, fully resolve your concerns, and encourage you to adjust your assessment accordingly.

---

### Note · Authors · 2025-08-11

This work introduces PartEvo, a Niching-enhanced LES (LLM-assisted Evolutionary Search) framework for efficiently Automated Algorithm Discovery (AAD). PartEvo represents **the first successful attempt** to integrate advanced Evolutionary Computation (EC) techniques, such as niching, within the LES paradigm. This addresses a core challenge: adapting EC techniques to the unique, abstract language space of LLMs, which is a significant departure from traditional numerical or genetic representations in traditional EC. Our comprehensive experiments demonstrate that PartEvo significantly improves AAD's efficiency, performance, and robustness compared to state-of-the-art LES methods like Eoh, providing a critical methodological paradigm for the future integration of a broader range of advanced EC techniques with LES.

During the rebuttal period, we made every effort to address all reviewer comments by undertaking substantial additional work and providing detailed explanations. We are confident we have not overlooked any of the reviewers' concerns. Our efforts included:
1. **Expanding our baseline comparison** by including ReEvo for a more comprehensive evaluation.
2. **Validating PartEvo’s compatibility** with various clustering methods.
3. **Providing a detailed discussion on algorithm parameter selection**.
4. **Investigating the influence of LLM capabilities** on overall algorithm discovery performance.
5. **Adding specific examples** to effectively illustrate PartEvo and highlight its advantages.
6. Conducting a comprehensive literature-based qualitative analysis **comparing the LES paradigm with Genetic Programming and Deep Learning-based AAD methods** in terms of training efficiency, resource requirements, and performance.
7. Identifying a future direction: a systematic study comparing the LES paradigm with non-LLM AAD methods. We argue this is a valuable area for future exploration, but not essential to the main claims of this work.

Reviewers DVie, h8jf, and S91M noted that we successfully resolved most of their concerns. Reviewer pyQe acknowledged our contributions to enhancing the performance and efficiency of the LES but requested a discussion on non-LLM-driven AAD methods. Given the limited time available, we provided the most comprehensive literature-based qualitative analysis possible (as detailed in point 6), but did not receive a further response before the deadline. We sincerely hope our response has resolved Reviewer pyQe's concerns.

---

### Decision · Program_Chairs · 2025-09-17

**Decision:**

Accept (poster)

**Comment:**

## Summary

This paper studies the use of large language models for automated algorithm design through evolutionary search. The authors introduce a framework that extends language model-assisted evolutionary search with feature-based niche construction, allowing methods from evolutionary computation to be applied in abstract program spaces. Based on this framework, they propose PartEvo, which integrates collaborative niche search with specialized prompting strategies. The method is evaluated on synthetic benchmarks and real-world optimization tasks. Results show that PartEvo improves upon earlier approaches such as EoH and Funsearch, and it achieves strong gains on resource scheduling problems, where it produces competitive meta-heuristics at low design cost. The findings indicate that niche-based strategies can enhance the effectiveness and practicality of evolutionary search with language models.

## Decision

This paper addresses the timely and important topic of automated algorithm discovery using evolutionary algorithms and large language models (LLMs). The problem itself is both significant and impactful. The paper is well-written, easy to follow, and the motivation is clear, supported by strong experimental evidence. The proposed approach is novel, and the results consistently demonstrate PartEvo’s superior performance across multiple benchmark tasks, substantially outperforming both human-designed baselines and prior LES-based methods. While the reviewers raised several important concerns during the rebuttal stage, the authors responded effectively and addressed them convincingly. I recommend this paper for acceptance, with the suggestion that the authors incorporate the additional experiments discussed during the rebuttal into the final version of the paper.